# Sub-micro droplet reactors for green synthesis of Li$_3$VO$_4$ anode materials in lithium ion batteries

Ha Tran Huu [1], Ngoc Hung Vu [2,3], Hyunwoo Ha [4], Joonhee Moon[5], Hyun You Kim [4] & Won Bin Im [1]✉

The conventional solid-state reaction suffers from low diffusivity, high energy consumption, and uncontrolled morphology. These limitations are competed by the presence of water in solution route reaction. Herein, based on concept of combining above methods, we report a facile solid-state reaction conducted in water vapor at low temperature along with calcium doping for modifying lithium vanadate as anode material for lithium-ion batteries. The optimized material, delivers a superior specific capacity of 543.1, 477.1, and 337.2 mAh g$^{-1}$ after 200 and 1000 cycles at current densities of 100, 1000 and 4000 mA g$^{-1}$, respectively, which is attributed to the contribution of pseudocapacitance. In this work, we also use experimental and theoretical calculation to demonstrate that the enhancement of doped lithium vanadate is attributed to particles confinement of droplets in water vapor along with the surface and structure variation of calcium doping effect.

[1] Division of Materials Science and Engineering, Hanyang University, Seoul, Republic of Korea. [2] Faculty of Biotechnology, Chemistry and Environmental Engineering, Phenikaa University, Hanoi, Vietnam. [3] Phenikaa Research and Technology Institute, A&A Green Phoenix Group, Hanoi, Vietnam. [4] Department of Materials Science and Engineering, Chungnam National University, Daejeon, Korea. [5] Advanced Nano-Surface Research Group, Korea Basic Science Institute, Daejeon, Republic of Korea. ✉email: imwonbin@hanyang.ac.kr

Since the first commercial products of Sony in 1991, lithium-ion batteries (LIBs) and other potentially replaceable metal-ion storage sources such as $Na^+$ ion[1,2], $Mg^{2+}$ ion[3], and $Al^{3+}$ ion[4] batteries, have attracted much attention for developing systems with high energy density, low cost, environmental benignity, and high safety[5]. Among the various methods investigated for the synthesis methods of LIBs electrode, the solid-state reaction (SSR) is a popular route as it is simple, does not require any solvent, and can be easily scaled up to the industrial level. However, this method does have several weaknesses. First, due to the low ionic diffusion in the solid state, SSR kinetics at room temperature are so low that the reaction cannot occur even when the ambient condition is thermodynamically favorable. Therefore, SSR synthesis always requires high temperature for long periods of time which implies huge energy consumption. Second, because the reaction can occur only at the solid/solid or gas/solid interfaces, the core may remain unreacted. Thus, it is difficult to achieve high uniformity and unwanted intermediate phases may be formed. Finally, treatment at high temperatures for a long time could lead to the agglomeration of particles with uncontrollable morphologies. However, the particle size is still so large that it could lead to poor electrochemical properties in the intercalated materials. Hence, it is necessary to investigate alternative strategies that satisfy these economical and environmental requirements, a strategy has been developed to overcome limitations of SSR[6,7].

In particular, the acid–base reaction (ABR) based process, which is carried out in the solid state in the humid atmosphere at low temperatures (below the boiling point of water). The addition of water vapor, formed by the evaporation of water at 80 °C, is the key factor in this modified pathway. In addition, the droplets formed during water condensation can not only serve as sub-micro droplet reactor in which the main ABR performed, but also confine the particle size and control morphology of the final materials. Based on this concept, we synthesized numerous materials as electrodes for LIBs or sodium-ion batteries (SIBs). In this investigation, a modification strategy for one such material, $Li_3VO_4$ is described. Inspire of its inherent advantages, such as a higher theoretical capacity (591 mAh g$^{-1}$) than graphite, low and safe voltage plateaus, and low cost, the actual utilization of $Li_3VO_4$ is restricted by drawbacks including large particle size and low electronic conductivity[8,9]. Therefore, many studies have been undertaken to overcome the aforementioned limitations using two strategies: (i) increasing its electronic conductivity (via doping[10–13], composite fabrication with graphene[14], carbon nanotube[15], or carbon coating[16], etc.) and (ii) reducing the particle size and controlling morphology. Compared to morphology engineering or composite fabrication, aliovalent substitution strategy was demonstrated as a direct and effective way to modify the electronic structure leading to enhanced electronic conductivity. Dong et al.[10] has reported that $Mo^{6+}$ doping to $V^{5+}$ could alter electronic band structure of $Li_3VO_4$ as n-type semiconductor and shift Fermi level toward conduction band due to induced extra electrons. Meanwhile, Ni-doped $Li_3VO_4$ with an improved surface energy could accelerate the insertion/extraction of $Li^+$ ions[13]. Besides, substitution of $Li^+$ by $Na^+$ could enhance electrochemical performance of $Li_3VO_4$ due to lattice parameter enlargement and particles size reduction[17]. Furthermore, the $Mg^{2+}$ introduction to $Li^+$ sites not only lead to lattice expansion but also enhance electronic conductivity leading to improvement in electrochemical properties. Therefore, $Ca^{2+}$ which is same group of $Mg^{2+}$ with larger ionic radius could be a good candidate for doping to $Li_3VO_4$.

In this study, we demonstrate the application of ABR to fabricate $Li_3VO_4$ and control its morphology and particle size. In addition, a green combination of the ABR strategy and Ca doping is employed to enhance the electrochemical properties of $Li_3VO_4$. Otherwise, a reaction mechanism is proposed to estimate the confined size of the droplet reactors in nanoscale and illuminate the doping effect on modification surface area of doped samples.

## Results

**Physicochemical characterizations.** As shown in X-ray diffraction (XRD) of Fig. 1a, at low contents of calcium (1 and 3%), the XRD profiles indicate a single orthorhombic phase of $Li_3VO_4$, indicating the successful substitution of $Ca^{2+}$ into $Li^+$ sites without the formation of an impurity phase. Nevertheless, at 5% Ca doping, the appearance of an unexpected peak, at 2θ ~30.67° (plus mark), indicates the formation of a new phase, $Ca_7V_4O_{17}$, in small amount. Meanwhile, the substitution of larger radius ion at

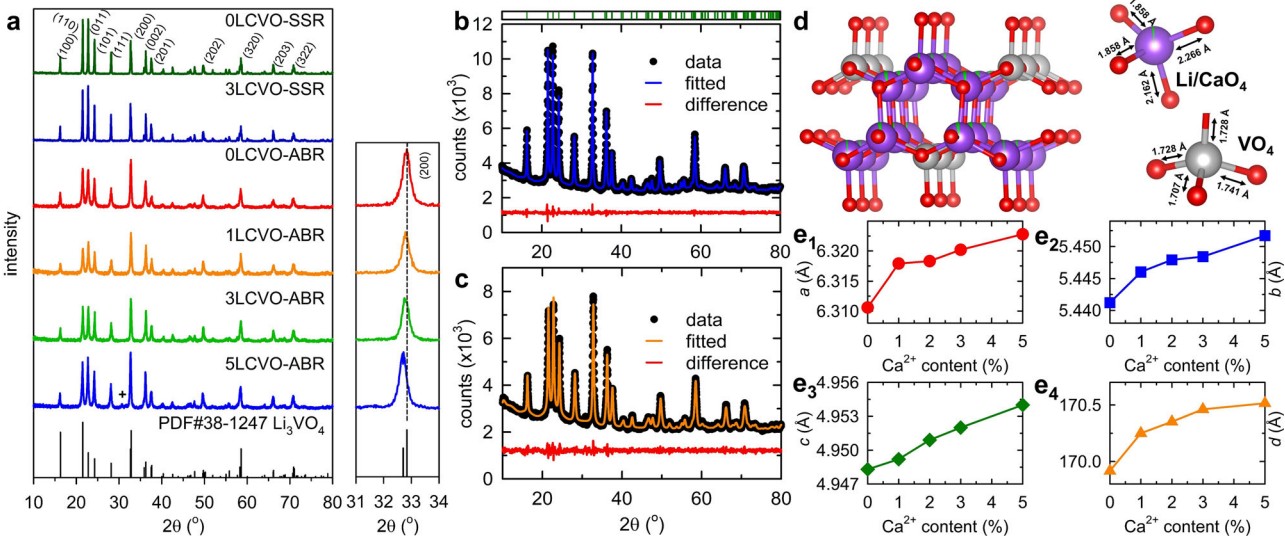

**Fig. 1 Structural analysis. a** XRD pattern of xLCVO-ABR and –SSR (x = 0, 1, 3, 5) (inset: expansion of XRD pattern at 2θ = 31–34°); Rietveld refinement based on HR-XRD of **b** 0LCVO-ABR and **c** 3LCVO-ABR; **d** Crystal structure of Ca-doped $Li_3VO_4$ at 3% of dopant: lithium, vanadium, and oxygen atoms were presented by purple, gray and red ball while Ca occupancy was presented by green contribution; dependence of lattice parameters **e₁** a, **e₂** b, **e₃** c and **e₄** unit cell volume on the content of Ca dopant.

the Li site could lead to the expansion of inter-planar $d$ spacing as illustrated by the shift of (200) peak to lower $2\theta$ angles (expansion of Fig. 1a)[11]. Rietveld refinement results performed on as-prepared samples, $x$LCVO-ABR ($x = 0, 1, 2, 3, 5$ equals to %Ca introduced in samples) are displayed in Fig. 1b, c, and Supplementary Figs. 1–3; the calculated lattice parameters are shown in Table 1, and Supplementary Table 1 while the structural parameters are listed in Table 2 and Supplementary Tables 2–4. An illustration of the crystal structure of the orthorhombic phase with a space group of $Pnm2_1$ constructing from two types of tetrahedral sites of $LiO_4$ and $VO_4$, is shown in Fig. 1d. It is evident that $Ca^{2+}$ ions prefer to replace $Li^+$ ions at $2a$ and $4b$ sites rather than V site owing to the fact that the ionic radius of $Ca^{2+}$ (1.0 Å, coordination number, CN = 6) is much larger than that of $V^{5+}$ (0.355 Å, CN = 4) but closer to that of $Li^+$ (0.59 Å, CN = 4). Moreover, the disparity in valence between $V^{5+}$ and $Ca^{2+}$ is larger than that between $Li^+$ and $Ca^{2+}$. Therefore, it is more propitious for $Ca^{2+}$ ions to occupy Li sites to form a non-impurity phase[10]. Furthermore, the data in Table 1 and Fig. 1e indicates that the $a$, $b$, and $c$ values and unit cell volume of Ca-doped samples increase slightly as the $Ca^{2+}$ dopant increases, thus confirming our conclusion of crystalline lattice enlargement, which is beneficial for enhancing $Li^+$ ion flexibility and rate capacity[10,17]. The formation of oxygen vacancies ($V_O$) to accommodate lattice strain due to the inconsistent of CN between $Ca^{2+}$ and $Li^+$ species can be determined by DFT calculation; such $V_O$ formation enhances ionic diffusion in doped samples.

For morphology investigation, the field-emission scanning electron microscopy (FESEM) image of 0LCVO-SSR (Supplementary Fig. 4a) exhibits a giant particle with dimension greater than 5 μm with a smooth morphology, caused by particle aggregation during sintering at high temperatures[17]. However, after doping with a small amount of Ca, 3LCVO-SSR (Supplementary Fig. 4b) exhibited much smaller particles with rough surfaces. The reduction in the particle size of the doped samples could be clarified by the variation in the surface energy or lattice

strain, which limits the crystal from further growth due to the occupation of hetero-atoms[18,19]. Meanwhile, in the ABR case, the SEM image in Fig. 2a illustrates the aggregation of rough and variegated particles. The few-ten-nanometer dimension demonstrates that the particle size was significantly reduced in the ABR method. Ca-doped samples obtained via the same reaction exhibit a similar morphology (as shown in Fig. 2b). However, the presence of $Ca^{2+}$ ions significantly altered the surface, as observed by the formation of open pore and a strongly non-uniform arrangement of primary particles (as shown in the transmission electron microscopy (TEM) images in Fig. 2c). While the edge surface of 0LCVO-ABR (Supplementary Fig. 4d) was tightly constructed with very less pores, the surface of 3LCVO-ABR (Fig. 2c) clearly exhibited a high porosity with a loose stacking of nanoparticles. In addition, the high-resolution TEM (HR-TEM, Fig. 2d) images show that the d spacing of the (100) plane of 3LCVO-ABR is 5.61 Å, which is larger than the theoretical value of 5.44 Å. This could be attributed to the larger ionic radius of $Ca^{2+}$ ion than that of $Li^+$ ion which is consistent with our XRD observation. The TEM elemental mapping shown in Supplementary Fig. 5 could confirm the well-distribution of Ca dopant. Further crystal lattice information was confirmed by selected area electron diffraction (SAED, Fig. 2e). In addition, elemental composition of as-prepared samples was investigated using Inductively Coupled Plasma Optical Emission Spectroscopy and X-ray fluorescence and the obtained results were summarized in Supplementary Table 5, which confirms the purity of synthesized samples.

To clarify the mechanism of particle size control via ABR, in situ Raman analysis was conducted during the preparation of pristine $Li_3VO_4$. The obtained data are shown in Supplementary Fig. 6, in which images in the first and second lines show photographs of the sample surface under green and white light. When observed by the naked eye (Supplementary Fig. 6a), it is clear that the initial LiOH and formed $Li_3VO_4$ are white and crystalline, and the yellow particles can be attributed to $V_2O_5$. Under green laser light (Supplementary Fig. 6b), however, the color of the substances changed distinctly with LiOH turning black, while $V_2O_5$ reflected green light and the product could be detected in the blue region. For clarifying a reasonable mechanism, in situ Raman analysis was conducted on three positions at specific time intervals throughout the reaction period. The initial position 1 is ascribed to $V_2O_5$, position 2 is the location of LiOH particles, and the remaining is the contact region between the two species. At the beginning of the reaction, LiOH was characterized by an intense peak located at 1091.6 cm$^{-1}$ [20] in the spectrum generated at position 2. Meanwhile, the highest intensity Raman peak of position 1 centered at 991.3 cm$^{-1}$ could be assigned to stretching mode ($A_{1g}$ and $B_{2g}$) of vanadyl group V=O. A low-intensity signal located at 951.3 cm$^{-1}$ is attributed to the asymmetric stretching ($B_{2g}$) vibration of the bridging bond V–O$_{(3)}$–V while the rail of the ladder of bridging

**Table 1 The refinement XRD information of 0LCVO-ABR and 3LCVO-ABR.**

| Sample | 0LCVO-ABR | 3LCVO-ABR |
|---|---|---|
| Symmetry | Orthorhombic | |
| Space grpoup | $Pmn2_1$ | |
| $a$ (Å) | 6.3106 (3) | 6.3183 (5) |
| $b$ (Å) | 5.4412 (2) | 5.4479 (4) |
| $c$ (Å) | 4.9483 (2) | 4.9509 (2) |
| $V$ (Å$^3$) | 169.916 (2) | 170.250 (3) |
| $\chi^2$ | 1.724 | 1.639 |
| $R_{wp}$ (%) | 2.50 | 2.58 |
| $R_p$ (%) | 1.95 | 2.00 |

**Table 2 Structural parameters of 3LCVO-ABR as obtained from the combined Rietveld refinement of X-ray.**

| Atom | Wyckoff position | x | y | z | 100 × $U_{iso}$ (Å$^2$) | g |
|---|---|---|---|---|---|---|
| O1 | $4b$ | 0.2269 (7) | 0.6833 (12) | 0.9098 (27) | 0.114 (2) | 1.0 |
| O2 | $2a$ | 0.0 | 0.1148 (17) | 0.9096 (29) | 0.115 (1) | 1.0 |
| O3 | $2a$ | 1/2 | 0.1801 (18) | 0.849 (4) | 0.372 (1) | 1.0 |
| Li1 | $4b$ | 0.2461 (2) | 0.3737 (5) | 0.9496 (8) | 0.556 (3) | 0.99 |
| Li2 | $2a$ | 1/2 | 0.8830 (8) | 0.9707 (6) | 0.155 (2) | 0.965 |
| V1 | $2a$ | 0.0 | 0.8330 (5) | 0.0134 (5) | 0.951 (5) | 1.0 |
| Ca1 | $4b$ | 0.2461 (2) | 0.373735 | 0.9496 (8) | 0.900 (1) | 0.025 |
| Ca2 | $2a$ | 1/2 | 0.8830 (8) | 0.9707 (6) | 0.800 (2) | 0.03 |

The numbers in parentheses are the estimated standard deviations of the last significant figure.

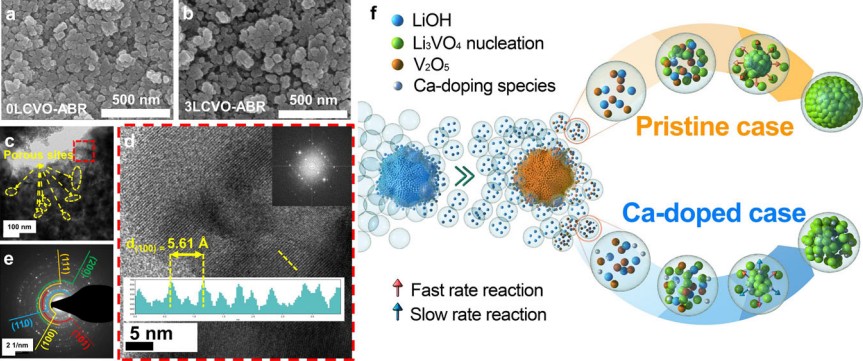

**Fig. 2 Morphology characterization.** SEM images of **a** 0LCVO-ABR and **b** 3LCVO-ABR; **c** TEM; **d** HR-TEM images, and **e** SAED pattern of 3LCVO-ABR, the FFT image for square inset and the yellow line profile for rectangle inset; **f** the proposal mechanism of ABR process and the effect of Ca doping on the morphology modification of 3LCVO-ABR.

V–O$_{(2)}$–V bonds with out-of-phase stretching vibration (B$_{1g}$ and B$_{3g}$) could be assigned for a peak at 702.5 cm$^{-1}$. The signal at 528.3 cm$^{-1}$ could be ascribed to the stretching mode (A$_{1g}$) of V–O$_{(2)}$ ladder steps. The V–O$_{(3)}$–V bridge with symmetric stretching vibration (A$_{1g}$) could be observed at Raman shift of 482.7 cm$^{-1}$ while a peak at 403.2 cm$^{-1}$ could accounted for the bending mode (A$_{1g}$) of this bridging bonds. The contiguous peaks at 292.1 and 281.4 cm$^{-1}$ could be assigned to the ladder-like distortion which is a combination of the in-phase vibration of V and O$_{(2)}$ atoms and the out-of-phase oscillation of O$_{(3)}$ atoms along the *b*-axis. The ladder breathing mode and the V=O$_{(1)}$ swinging vibration could be attributed to the signal at 201.1 cm$^{-1}$ while the lowest Raman shift signal at 147.6 cm$^{-1}$ (B$_{1g}$) could be accounted for the shearing-like distortion which is originated from the symmetric vibration of V, O$_{(2)}$, and O$_{(3)}$ atoms along the *b*-direction[21,22]. After 90 min of reaction, the peaks' intensity at position 1 decreased and this effect might be attributed to the surrounding of water layer as well as dissolving consumption by droplet. Whereas, the spectrum of position 2, beside peaks related to LiOH, new peaks in the range of 250–500 and 750–1000 cm$^{-1}$ corresponds to the formation of Li$_3$VO$_4$, indicating the higher reaction rate of these areas which can be ascribed to the high solubility of LiOH[23–25]. Similar signals could be observed in the spectrum at position 3, which is somewhere with well-mixed precursors particles, demonstrating that the formation of Li$_3$VO$_4$ could occur at any delocalized position due to the mobility of the droplet reactor. In addition, the reappearance of the peak at 1091 cm$^{-1}$ in Raman spectrum of position 3 after 360 min could prove the mobility of droplet reactors containing LiOH precursor. The reduction of LiOH-related peaks at ~1091 cm$^{-1}$ and the main contribution of Li$_3$VO$_4$ signal after 270 min of reaction could demonstrate for the effect of mobile vapor droplets on difference of reaction rate. The low solubility of V$_2$O$_5$ in water inhibits the synthesis reaction on site of this precursor, which is confirmed by the spectrum at position 1, in which only the peaks of V-precursor were remaining. In contrast, in other places, the formation of the final product was almost complete after 270 min. Based on this data, a hypothesized mechanism may be proposed for the ABR. As simulated in Fig. 2f, first, the precursor particles are surrounded by water molecules, and hence, they exhibit different behaviors depending on their solubility. Owing to the low water solubility of V$_2$O$_5$, water vapor is only absorbed on these particles and make their surfaces more acidic. In contrast, LiOH, which exhibits high solubility in water, may be diluted in water droplets and easily transferred to other places, especially acidic V-surfaces, to perform the main ABR. After moving to the surface of V$_2$O$_5$, the basic environment of the LiOH droplet dissolves the vanadium precursor to form nucleating Li$_3$VO$_4$ species, which are more

water soluble. At the limit of solubility of Li$_3$VO$_4$, they crystallize to form solid Li$_3$VO$_4$ while water evaporates back to begin another cycle. However, the size of particles growing on nucleating groups is confined by water vapor droplets to several tens of nanometers. The detailed calculation of water droplet size is presented in Supplementary Discussion. Accordingly, the maximum radius to which the droplet can grow can be calculated using Supplementary Eq. 1, according to which, at a saturation degree (S) of 0.872, the maximum radius to which the droplet can grow is in the range of 25.56 nm (for a total number of moles of solute $N_s = 10^{-18}$ mol) to 2.58 μm ($N_s = 10^{-15}$ mol). To confirm our results, we calculate the amount of Li$_3$VO$_4$ dissolved in a droplets as follows. Nanosized particles of Li$_3$VO$_4$ were assumed to be spherical shape with an approximate radius of 25 nm and mass density of $2.47 \times 10^3$ g m$^{-3}$. It is easy to derive the solute concentration as $1.18 \times 10^{-18}$ moles which is consistent with the initial value. The calculated value is in agreement with a previous report on the droplet size distribution of the water vapor system[26,27]. As the vapor droplets act as sub-micro reactors, the primary particles are limited not only to growth but also condense together to form the final morphology, as shown in the FESEM image.

In addition, the linear fitted Brunauer–Emmett–Teller (BET) results, (Supplementary Fig. 7), present the specific surface areas of all the samples, as summarized in Table 3. In detail, the two samples prepared by ABR possess a higher surface area than the corresponding SSR samples, with 3LCVO-ABR exhibiting the maximum surface area of 4.484 m$^2$ g$^{-1}$ (3.2 and 29.5 times higher than those of 0LCVO-ABR and 0LCVO-SSR, respectively). The *t*-plot in Supplementary Fig. 8 indicates that ABR synthesis and Ca doping tend to increase the μ-pore area, while the pore size distribution in Supplementary Fig. 9 implies that the pore size was higher in 3LCVO-ABR when compared to the pure sample. These results demonstrate that the combination of ABR and Ca doping induces a mesoporous morphology and significantly increases the specific surface area. This in turn yields a strong interface between the active materials and electrolyte, leading to a higher reaction area and shortening of the Li$^+$ ion movement pathway[13,28,29]. Such increase in the surface area can be explained as follows. When doping Ca, the foreign atoms act as a modifying agent, which changes not only the surface energy but also the mobility and reaction kinetics as they exhibit different hydration energy and diffusivity when compared to Li$^+$ ions. First, due to the more negative hydration enthalpy of Ca$^{2+}$ (−1577 kJ mol$^{-1}$) as compared to Li$^+$ (−520 kJ mol$^{-1}$), the absorbed Ca$^{2+}$ could lower the reaction surface energy by enhancing the lattice surface tension of Li$_3$VO$_4$[30]. This energy also reflects the fact that the stronger binding of Ca sites with water molecules inhibits further

**Table 3 Calculation of BET surface area.**

| Sample | BET linear plot | Total surface area ($m^2 g^{-1}$) | External surface area ($m^2 g^{-1}$) | Total pore volume ($cm^3 g^{-1}$) |
|---|---|---|---|---|
| 0LCVO-SSR | $y = 28.36x + 0.280$ | 0.152 | 0.1433 | 0.00129 |
| 0LCVO-ABR | $y = 3.07x + 0.042$ | 1.400 | 1.3100 | 0.01037 |
| 3LCVO-SSR | $y = 4.26x + 0.023$ | 1.024 | 0.9914 | 0.01136 |
| 3LCVO-ABR | $y = 0.96x + 0.005$ | 4.484 | 4.0860 | 0.05519 |
| 3LMVO-ABR | $y = 0.62x + 0.004$ | 6.963 | 6.1260 | 0.07040 |

crystallization. Second, as water droplets act as sub-micro reactors, the main parameter determining the reaction rate is reactant diffusivity. As shown in Eqs. 1 and 2, the reaction rate generally depends not only on the intrinsic reaction rate but also on the rate at which molecules diffuse close to each other; this effect is called the diffusion-influenced reaction:

$$A + B \underset{k_{-d}}{\overset{k_d}{\rightleftharpoons}} \{AB\} \overset{k_r}{\longrightarrow} \text{product(s)} \tag{1}$$

$$\frac{d[\text{product}]}{dt} = k_r \frac{k_d}{k_{-d} + k_r}[A][B] = k[A][B] \tag{2}$$

where, $k_d$ and $k_{-d}$ are reversible diffusion coefficients and $k_r$ is the intrinsic reaction rate. According to Eq. 2, the diffusivity of $Ca^{2+}$ ions ($7.93 \times 10^{-10}$ $m^2 s^{-1}$) is lower than that of $Li^+$ ions ($10.3 \times 10^{-10}$ $m^2 s^{-1}$) due to its larger size and higher valence[31]. Therefore, the reaction at the Ca site could be slower than further crystal growth at a Li site. For the two reasons mentioned above, $Ca^{2+}$ ions in $Li_3VO_4$ act as inhibitors, leading to a non-uniform surface with high porosity. This enhanced porosity is indicated by the significantly higher surface area of 3LCVO-ABR as compared to the remaining samples. To further confirm the effect of $Ca^{2+}$ ions on surface area modification, Ca was replaced by magnesium ($Mg^{2+}$) (at the same content), which has a higher hydration enthalpy ($-1926$ $kJ mol^{-1}$) and lower diffusivity ($7.06 \times 10^{-10}$ $m^2 s^{-1}$). The XRD pattern of the Mg-doped sample in Supplementary Fig. 10 can be used to determine the purity of the prepared sample while the increase in BET surface area, large μ-pore area, and large pore size of 3LMVO-ABR confirm the inhibitory effect of the dopant ions on particles growth, as demonstrated in Supplementary Figs. 7–9.

Subsequently, the chemical valence state of the elements present in the as-prepared samples was analyzed by X-ray photoelectron spectroscopy (XPS), as shown in Supplementary Fig. 11a. The $V2p$ spectra of 0LCVO-ABR, 3LCVO-ABR, and 3LCVO-SSR included $V^{5+}2p_{3/2}$ and $V^{5+}2p_{1/2}$ at binding energies of ~517.0 and 525.5 eV, respectively. In particular, the $V2p_{3/2}$ peak could be deconvoluted into two peaks; for example, in 3LCVO-ABR, the $V2p_{3/2}$ peak could be deconvoluted into peaks at 517.62 and 525.02 eV (spin orbit splitting energy of 7.48 eV) corresponding to $V^{5+}$; 516.77 and 524.15 eV (spin orbit splitting energy of 7.38) accounted for $V^{4+}$, respectively[32]. The estimated ratio of $V^{4+}/V^{5+}$ in the doped samples was 34.52% which is 5.21 times higher than that of pure material (6.63%). In addition, the co-existence of pentavalent and tetravalent vanadium species could be explained by the compensation the unbalanced charges caused by the substitution of $Li^+$ with ions of higher valence according to the Knöger–Vink equation:

$$2Li_{Li}^{\times} \rightarrow Li_{Ca}^{\bullet} + V_{Li}' \tag{3}$$

$$\text{or } Li_{Li}^{\times} \rightarrow Li_{Ca}^{\bullet} + e' \tag{4}$$

in which excess electrons result in the reduction of $V^{5+}$ to $V^{4+}$ due to the oxygen corner-sharing location of $LiO_4$ and $VO_4$ tetrahedrons.

$$V^{5+} + e' \rightarrow V^{4+} \tag{5}$$

The additional formation of low-valence V species[33], causes further enlargement in the lattice crystal due to the larger ionic radius of $V^{4+}$ compared to $V^{5+}$[11,29,34]. Furthermore, due to the increase in the content of low-valence species, $V_O$ related to $V^{4+}$ is reinforced, which leads to the disappearance of corner-sharing between $LiO_4$ and $VO_4$ tetrahedrons which lead to more space for $Li^+$ ions to diffuse[34]. As shown in Supplementary Fig. 11b, the $O1s$ spectra of as-prepared samples could be distinguished into three components (for 3LCVO-ABR) such as the highest intensity peak at 530.26 eV corresponding to oxygen bonding with lithium and vanadium in corner-sharing tetrahedrons and two weak constituents at 531.93 and 533.15 eV related to oxygen defect and absorbed oxygen on the surface, respectively[8,17,35,36]. From the Supplementary Fig. 11b, it can be calculated that the concentration of $V_O$ in 3LCVO-ABR increased to 30.4% when compared to 6.8% in the pristine sample, which is proportional to the $V^{4+}$ contents. Meanwhile, 3LCVO-SSR exhibited a oxygen defect content as 9.94% which indicates that a protective layer created by water vapor at low temperatures are advantageous for $V_O$ formation. Conversely, by treating at high temperatures in air, $V_O$ can be partly filled and suppressed by excess ambient oxygen molecules thus reducing the $V_O$ concentration to one-third of the value in doped ABR samples. The presence of lattice defects and the single electron configuration of $3d^1$ in $V^{4+}$ was confirmed by electron spin resonance (ESR, Supplementary Fig. 12).

To confirm the formation of oxygen vacancies and increase in $V^{4+}$ species induced by Ca doping, DFT calculations were conducted. It was found that the Ca dopants reduced the formation energy of oxygen vacancy, $E_{vac}$, in $Li_3VO_4$ (Supplementary Table 6). The average $E_{vac}$ of three types of oxygen ions (O1, O2, and O3) in $Li_3VO_4$ (3.99 eV) was slightly decreased to 3.50 eV upon Ca doping. However, the $E_{vac}$ of specific oxygen species (O1, highlighted in blue in Fig. 3c), which was composed of a tetrahedron with the V ion at the central site between two Ca dopants (highlighted in yellow in Fig. 3d), decreased from 4.06 to 3.33 eV. This finding indicates that the effect of Ca doping is highly localized and that extra oxygen defects can be formed around the dopants. Bader charge analysis shows that the $V^{5+}$ ion located between two Ca dopants was reduced upon Ca doping (Fig. 3e). The electron density difference map estimated by abstracting the electron density of $Li_{48}V_{16}O_{64}$ from that of $Li_{46}Ca_2V_{16}O_{64}$ shows that the extra electrons donated by Ca dopants are localized to V and adjacent $O^{2-}$ ions (Fig. 3f). These results corroborate the dual-functionality of Ca dopants, viz. (1) directly donating electrons to $V^{5+}$ ions, and (2) accelerating the generation of oxygen vacancies which can further reduce $V^{5+}$ ions. Therefore, the effects of $Li^+$ substitution with higher valence ions, co-existence of $V^{4+}$ and $V^{5+}$ and $V_O$ formation are all related to the increase in excess free electron defects in the structure. This enhances the electronic conductivity of the material and reduces the polarization during the insertion and

extraction of $Li^+$ ions into or from $Li_3VO_4$ lattice, which positively impacts materials' cyclability and rate performances[33,37,38].

**Electrochemical properties.** The galvanostatic charge–discharge results of all the tested samples corresponding to the first cycle are presented in Fig. 4a. A typical discharge profile includes a sharp slope at voltages higher than 0.8 V attributed to lithium ion insertion into the lattice and a smooth region with two characteristic plateaus at ~ 0.75 and 0.6 V indicative of the interaction between active materials and the electrolyte, leading to the formation of a solid electrolyte interphase (SEI) layer in the first few cycles[16,39]. Although all samples exhibited similar curves, the initial discharge capacity and first coulombic efficiency (CE) of the doped and ABR samples showed obvious enhancement (Supplementary Fig. 13). The initial charge/discharge capacities of pure SSR and ABR samples were quite similar at 263.43/441.79 and 305.4/568.91 mAh g$^{-1}$ with a CE of 59.63% and 53.68%, respectively. However, the first discharge capacity of 3LCVO-ABR was as high as 946.78 mAh g$^{-1}$ with a very high CE of 74.19%. The huge loss of capacity in the few first cycles could be ascribed to irreversible lithium ion loss due to side reactions, including formation of SEI layer and electrolyte decomposition[40,41]. The apparent increase in the capacity of doped ABR sample could be explained by the effect of particle size and specific surface area[42]. The nanosized particles and higher surface area in ABR samples are favorable for improving electron exchange and reaction surface contact, which contribute to their high capacity. In the case of 3LCVO-ABR, its high surface area might explain for its highest capacity, while the enhanced CE value might be associated with the

increase in electronic conductivity due to Ca doping[33,37,38]. The substitution of $Li^+$ with higher valence ions not only generates excess electrons but also change the electronic band structure, leading to the shift of Fermi level toward the conduction band (due to the reduction of $V^{5+}$)[37,38]. Furthermore, another important reason related to the low first CE of $Li_3VO_4$ could be ascribed to crystallite distortion due to the large amount of Li ions inserted or extracted during the first charge/discharge process[39]. The higher the number of Li ions inserted to form $Li_{3+x}VO_4$ (especially $x > 2$), the more monoclinic (even triclinic, at $x = 3$) is the distortion from the original orthorhombic phase. The higher CE of 3LCVO samples compared to the pure samples demonstrates the positive effect of Ca doping in enhancing the structural flexibility under deeply lithium exchange. According to the in situ XRD shown in Supplementary Fig. 14, while lithiation/delithiation in the first cycle of 0LCVO-ABR perform a formation of unknown peak (phase II) related to distortion of pristine structure to the secondary phase which is considered as main cause for the irreversible capacity loss in the first cycle[43], there is no new phase could be observed in the XRD pattern of 3LCVO electrode. This result could illustrate for the effect of Ca doping on regulating the lattice structure of $Li_3VO_4$ toward higher adaptability of lithium ions insertion/extraction.

In addition, it can be observed that at a current density of 100 mA g$^{-1}$, the cycling performance of the doped samples prepared by ABR was enhanced. The fading capacity of all samples in the few first cycles could be regarded as the initial activation required to lower the insertion barrier by irreversible phase transition[39]. However, the ABR samples exhibited the superior capacity retention of 92.72%, 94.27%, and 89.72% after 200 cycles for

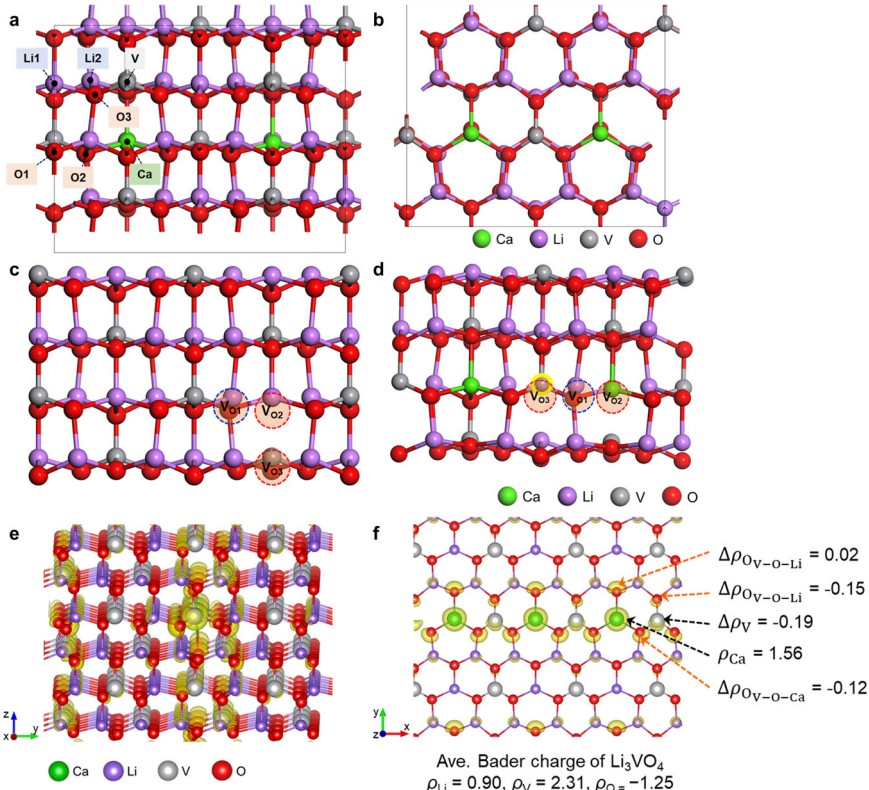

**Fig. 3 Density functional theory calculation.** Morphology of the $Li_{46}Ca_2V_{16}O_{64}$ supercell via **a** (010), and **b** (001) plane direction; location of oxygen species used for $E_{vac}$ calculation: **c** $Li_{48}V_{16}O_{64}$ and **d** $Li_{46}Ca_2V_{16}O_{64}$. Electron density analysis results. **e** The electron density difference map ($\rho = 0.05$ e/Å$^3$) estimated by abstracting the electron density of $Li_{48}V_{16}O_{64}$ from that of $Li_{46}Ca_2V_{16}O_{64}$. The yellow highlighted shades present at which extra electrons are localized. **f** The sliced single atomic layer of $Li_{46}Ca_2V_{16}O_{64}$ involving the Ca dopants. The electron charge density map and the Bader charge analysis results are accordingly presented. The $\Delta\rho$ denotes the Bader charge difference. The ions with negative $\Delta\rho$ are enriched with extra electrons.

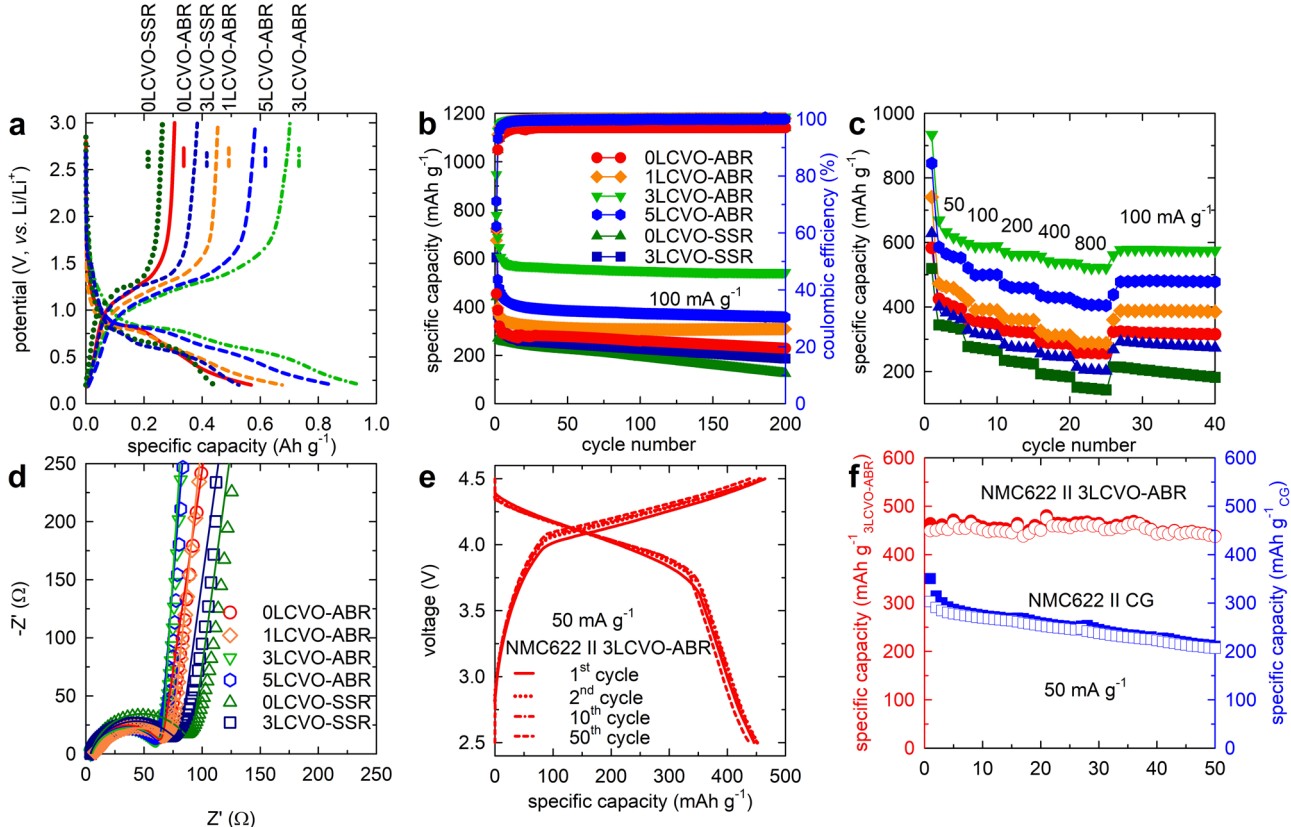

**Fig. 4 Electrochemical properties of half- and full cells. a** the first galvanostatic charge–discharge curves **b** cycling performance and **c** c-rate behavior **d** EIS fitting analysis of xLCVO-ABR and -SSR (x = 0, 1, 5); **e** galvanostatic charge–discharge curve and **f** cycling performance of full cell of NMC622 cathode and 3LCVO-ABR anode or CG.

xLCVO-ABR (x = 1, 3, 5, respectively) and 90.27% for 0LCVO-ABR. The decrease in capacity retention beyond doping concentration of 3% can be explained by the lithium-ion diffusion blocking effect at a high level of doping. However, the 0LCVO-SSR and 3LCVO-SSR samples could retain only 55.76% and 76.09% after 200 cycles for 0LCVO-SSR and 3LCVO-SSR, respectively. This means that the ABR samples not only showed greater cycling performance but also delivered higher reversible capacity, 543.1 mAh g$^{-1}$ (for 3LCVO-ABR) and 268.7 mAh g$^{-1}$ (for 0LCVO-ABR) after 200 cycles. Even at a higher current density of 1000 and 4000 mA g$^{-1}$ (Supplementary Fig. 15), 3LCVO-ABR still displayed a high reversible capacity of 477.1 and 337.2 mAh g$^{-1}$ and capacity retention of 91.7% and 90.6% after 1000 cycles. Meanwhile, the SSR samples revealed poor cyclability with a dramatic crash in capacity after only a few cycles. The high reversible capacity 3LCVO-ABR could be ascribed to the contribution of pseudocapacitive lithium ion storage pathway, which will be demonstrated in the following discussion.

Beside the excellent cycling performance, the rated performance at current densities in the range of 50–800 mA g$^{-1}$ of Ca-doped ABR samples (Fig. 4c) was improved. At a current density of 50 mA g$^{-1}$, the capacity of 3LCVO-ABR, was higher than 600 mAh g$^{-1}$, which is greater than theoretical specific capacity of Li$_3$VO$_4$ (591 mAh g$^{-1}$), attributed to the continuous formation of a jelly-like polymeric SEI layer due to the unstable nature of this layer to electrolyte[39,44,45]. In addition, at low potential, the interfacial storage mechanism[46,47] may also explain the extra capacity[48,49]. The excellent cycling behavior and rated performance of the ABR samples, especially 3LCVO-ABR, may be attributed to their enhanced electronic conductivity and Li$^+$ ion

diffusivity, as demonstrated by the electrochemical impedance spectroscopy (EIS) data in Fig. 4d (enlargement of low-frequency range in Supplementary Fig. 16) and Supplementary Fig. 17. All the obtained EIS curves included a semicircle at high frequencies corresponding to charge-transfer resistance and a linear incline in the low-frequency region corresponding to the Li$^+$ ion diffusivity in the electrodes. The results of EIS fitting, deduced using the equivalent circuit model as shown in Supplementary Fig. 18, are summarized in Table 4. It can be noted that the doped samples exhibited a lower charge-transfer resistance, which is consistent with our previous observation. Furthermore, the diffusion coefficient of lithium ions was calculated using the EIS data in the inclined region following the method presented in Supplementary Methods (Supplementary Fig. 17). The lowest slope of linear plot indicating the highest Li$^+$ ion diffusion coefficient obtained for 3LCVO-ABR illustrates that the mobility of Li ions was effectively accelerated by the expanding the lattice parameter and increasing the surface area of the electrode.

In order to clarified the effect of Ca doping on electronic and ionic conductivity of Li$_3$VO$_4$, a theoretical calculation has been conducted. Herein, we have conducted a calculation on the energy barrier of lithium-ion diffusion in two pathway: (i) parallel Li$^{2a}$–Li$^{2a}$ transport, and (ii) perpendicular Li$^{2a}$–Li$^{4b}$ hoping. Accordingly, the highest activation energies ($E_{act}$) of lithium-ion diffusion in Li$_{48}$V$_{18}$O$_{64}$ supercell are 0.36 eV (Li$^{2a}$–Li$^{2a}$ parallel, Supplementary Fig. 19a) and 0.25 eV (Li$^{2a}$–Li$^{4b}$ perpendicular, Supplementary Fig. 20a) which demonstrates that both directions are favorable for transport of lithium ions. However, the overall morphology of lithium-ion diffusion in a model of Li$_{46}$Ca$_2$V$_{18}$O$_{63}$ is a stepwise process within intermediate position stabilized by the presence of Ca substitution and oxygen vacancy. The increase

**Table 4 EIS Charge-transfer resistances, linear relation of Z′ versus $\omega^{-1/2}$, and lithium-ion diffusion coefficients of the samples.**

| Samples | Ohmic resistance (Ω) | Charge-transfer resistance (Ω) | σ (Ω s⁻⁰·⁵) | $D_{Li^+}$ (cm² s⁻¹) |
|---|---|---|---|---|
| 0LCVO-SSR | 4.22 | 87.81 | 69.206 | $6.4 \times 10^{-11}$ |
| 0LCVO-ABR | 7.90 | 71.65 | 41.368 | $1.07 \times 10^{-10}$ |
| 1LCVO-ABR | 4.35 | 62.34 | 101.397 | $4.4 \times 10^{-11}$ |
| 3LCVO-ABR | 3.64 | 57.64 | 41.057 | $1.08 \times 10^{-10}$ |
| 5LCVO-ABR | 2.85 | 67.69 | 46.729 | $9.4 \times 10^{-11}$ |
| 3LCVO-SSR | 3.88 | 78.98 | 111.900 | $3.9 \times 10^{-11}$ |

of the highest $E_{act}$ in $Li_{46}Ca_2V_{18}O_{63}$ model determined the sluggish effect of Ca doping on diffusion of lithium ions. However, the significant increase in energy barrier as shown in Supplementary Fig. 20a, b indicates the deactivation of perpendicular pathway in doped sample, while the slight raise of 0.06 eV in parallel movement is acceptable[50,51]. Therefore, a regulation of lithium-ion diffusion in Ca doping case into single parallel type could be presumable origin for the enhancement in ionic conductivity observed in experimental data. In addition, the enhancement in lithium-ion diffusion of doped samples is also originated from the crystallite size calculated by Williamson–Hall (WH) plot. As shown in Supplementary Fig. 21 and summarized in Supplementary Table 7, there is a significant decrease in crystallite size of 3LCVO-ABR compared to the remaining samples leading to an increase of grain boundary at which ionic conductivity is much higher than grain interior[52,53]. Therefore, a reduction in crystallite size along with particle size could accelerate the enhanced ionic conductivity of doped sample. Furthermore, the microstrain, which orginates the micro-crack and pulverization of particle, extracted from WH plot (Supplementary Table 7) exhibits a significant decrease in 3LCVO-ABR compared to pure samples. This less lattice mismatching observed after Ca doping is favorable for improving cycling performance of active materials[54]. Along with the stabilization of lithium ion after Ca doping observed in DFT calculation, the reduction of microstrain is consistent to the previous discussion on enhancement in structural flexibility, which originates the better first CE. In addition, the electronic band diagrams were present in Supplementary Fig. 22 and the information of electron occupation was summarized in Supplementary Tables 8–13. According to Supplementary Fig. 22, the valence band maximum (VBM) and conduction band minimum (CBM) are −0.7016 eV and 5.1450 eV, corresponding to a bandgap of 5.8466 eV. The introduction of $Ca^{2+}$ into lattice of $Li_{46}Ca_2V_{18}O_{64}$ induced a significant shift of both VBM and CBM toward a lower bandgap of 4.5061 eV. In addition, the excess electrons induced from charge compensation could occupy the energies level higher than VBM which means more probability for electron to excite to conduction band. The similar result was also observed in additional formation of oxygen vacancy. Furthermore, the reduction effect of these excess electrons could induce the formation of mixed oxidation state of $V^{4+}/V^{5+}$ which shifts the Fermi level toward conduction band. Therefore, besides the enhancement in ionic diffusion, the electronic conductivity improvement of doped samples could be attributed to the modification of Ca doping on bandgap structure.

The potential in the practical application of 3LCVO-ABR electrode could be evaluated using full cells, which were assembled from a $LiNi_{0.6}Mn_{0.2}Co_{0.2}O_2$ (denoted as NMC622) cathode and 3LCVO-ABR or commercial graphite (denoted as CG) anode. The 3LCVO-ABR electrode was cycled in a half cell for 20 cycles to reach the stable cycling period before applied in full-cell operation[55,56]. The galvanostatic profile of the NMC622‖3LCVO-ABR full-cell at a current density of 50 mA g⁻¹ was shown in

Fig. 4e. In the first cycle, the full-cell of 3LCVO-ABR delivered the specific discharge capacity of 462.8 mAh g⁻¹ and retain as 92.7% (429.1 mAh g⁻¹) after 50 cycles. Meanwhile, the NMC622‖CG full cell, only delivered a specific discharge capacity of 350.6 mAh g⁻¹ in its first cycle, and a capacity retention of 60.0% after 50 cycles, which corresponds to the capacity loss of 0.79% per a cycle. The lower in capacity of full-cell operation could be attributed to the cycling condition which should be optimized in the future[56,57]. However, with the recent results of the full cells, it is reliable to imply that 3LCVO-ABR could be a promising practical anode for LIBs. The brief comparison on electrochemical performance of state-of-the-art $Li_3VO_4$-based anode is also present in Table 14.

Figure 5a shows the cyclic voltammetry CV results of 3LCVO-ABR for four initial cycles, while the CV curves in the second cycle corresponding to pure and representative doped samples synthesized by ABR are shown in Supplementary Fig. 23b. Pristine ABR-$Li_3VO_4$ showed the 2nd cathodic peaks at 0.500 and 0.846 V, corresponding to the reduction of $V^{5+}$ to $V^{4+}$ and $V^{3+}$ species, respectively, and an anodic peak at ~1.168 V related to the reversible oxidation of $V^{3+}$ [12]. Meanwhile, the 3LCVO-ABR displayed three peaks at 0.60, 0.896, and 1.108 V. This means that based on the doping strategy, peaks corresponding to reduction tend to shift to higher voltage regions while the oxidation peak moves toward lower potentials, which reduces the voltage gaps from 0.668/0.322 V for pristine samples to 0.504/0.212 V in 3LCVO-ABR. The observed potential values corresponding to reduction and oxidation peaks are summarized in Table 5. The smaller voltage gap reconfirms our observation on reducing polarization in the electrode and accelerating lithium diffusion[10,58], which is beneficial for cycling and rate performance. The ex situ XPS (Supplementary Fig. 24) was conducted on 3LCVO-ABR electrode at initial state and after being discharged to 0.1 V and charged to 3.0 V, vs. Li/Li⁺. Accordingly, the V2p XPS of 3LCVO-ABR electrode at 0.1 V consists of three components accounted for $V^{3+}$ at 515.51 eV, $V^{4+}$ at 516.45 eV, and $V^{5+}$ at 517.42 eV. After charged to 3.0 V, vs. Li/Li⁺, the V2p XPS is only composed of $V^{4+}$ and $V^{5+}$ signals. These results are consistent to the observation in CV.

As aforementioned, an additional strategy for lithium ions which is corresponding to the capacitive surface reaction could be attributed as the origin of excess capacity observed in 3LCVO-ABR[59-63]. Due to the requirement of extremely large specific surface area (up to 2000 m² g⁻¹)[60,64], the electrical double layer capacitive mechanism is not reasonable to apply in case of 3LCVO-ABR. Therefore, besides the contribution of the diffusion-controlled process, the pseudocapacitane could be an additional contribution to the total charge storage of 3LCVO-ABR.

In order to demonstrate the presence of pseudocapacitance, the CV plots measured at scan rates varied from 0.1 to 5.0 mV s⁻¹ are conducted and presented in Fig. 5b. The detailed calculation is described in the Supplementary Methods. Accordingly, the calculated $b$-values decrease to 0.59 at potential of 1.2 V, vs. Li/Li⁺ (for cathodic process) or 0.66 at potential of 0.82 V, vs. Li/Li⁺ (for anodic process), which indicate the main contribution of intercalation reaction at

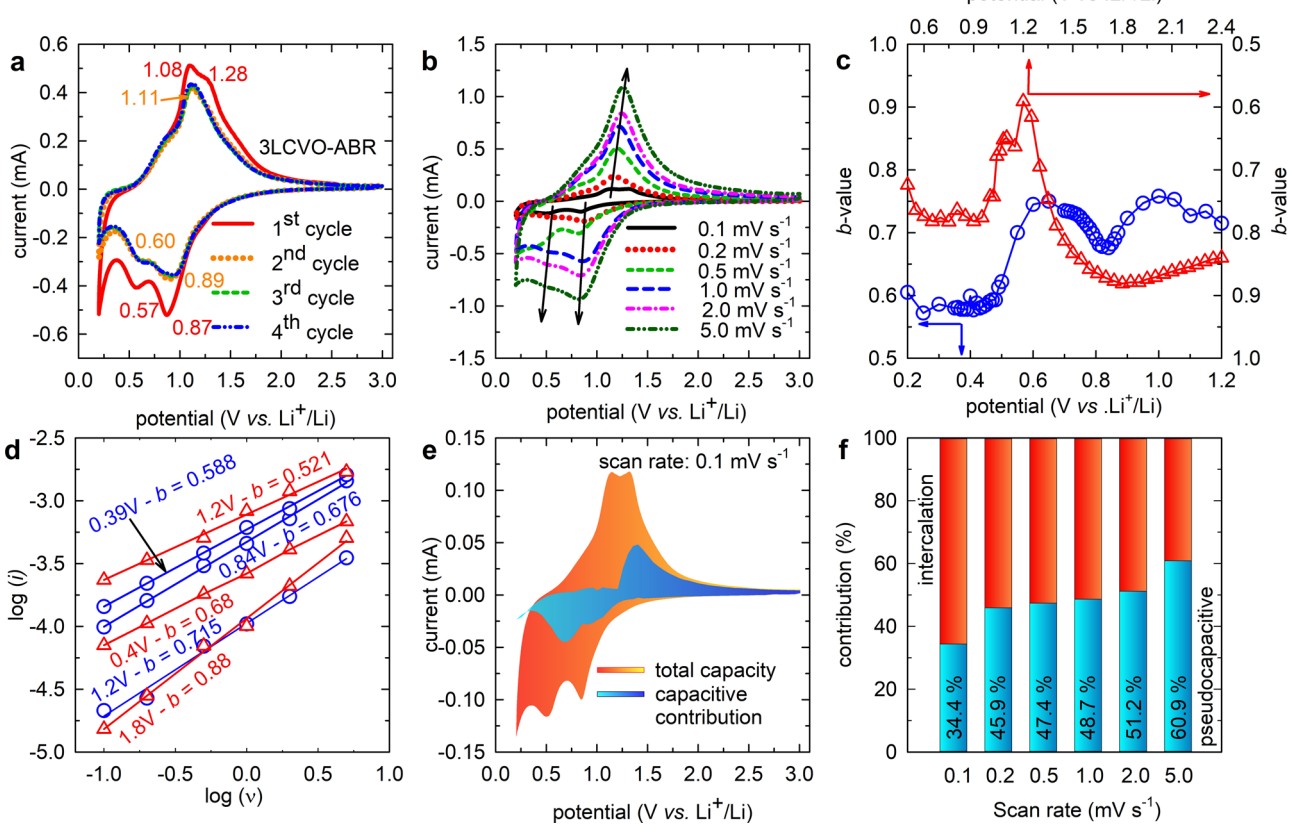

**Fig. 5 Pseudocapacitance analysis. a** CV of 3LCVO-ABR; **b** CV of 3LCVO-ABR at various scan rate $n = 0.1–5.0$ mV s$^{-1}$; **c** dependence of $b$-value on potential in a range of cathodic (red color) and anodic (blue color) peaks; **d** linear fitting of log(i) vs. log($\nu$) for determination of $b$-value; **e** pictorial estimation of pseudocapacitance contribution; and **f** the contribution of pseudocapacitive effects in the total charge storage of the 3LCVO-ABR.

**Table 5 The voltage differences between the oxidation and reduction peaks of the 0LCVO-ABR and 3LCVO-ABR.**

| Sample | Cycle | $\varphi_o$(V) | $\varphi_{r_1}$(V) | $\varphi_{r_2}$(V) | $\Delta\varphi_{r_1}$(V) | $\Delta\varphi_{r_2}$(V) |
|---|---|---|---|---|---|---|
| 0LCVO-ABR | 2nd | 1.17 | 0.50 | 0.86 | 0.67 | 0.31 |
| | 3rd | 1.15 | 0.51 | 0.85 | 0.64 | 0.30 |
| | 4th | 1.12 | 0.52 | 0.85 | 0.60 | 0.27 |
| 3LCVO-ABR | 2nd | 1.11 | 0.60 | 0.89 | 0.51 | 0.22 |
| | 3rd | 1.10 | 0.61 | 0.91 | 0.49 | 0.19 |
| | 4th | 1.09 | 0.62 | 0.92 | 0.47 | 0.17 |

doping strategies, which significantly enhances the electrochemical performance of Li$_3$VO$_4$ anodes for LIBs, is investigated to understand the reaction mechanism corresponding to ABR synthesis. The proposed mechanism, which illustrates the role of sub-micro reactors of vapor droplets, details the effect of nano-sized particles, while doping leads to increase in crystal lattice parameters, forms beneficial defects, and induces enlargement in specific surface area. The high reversible capacity and long-life cycling behavior are attributed to the synergic effects of modification in lattice structure, the favorable morphology which not only accelerates the diffusivity of charge carriers but also offers the extra contribution to final energy storage via a pseudocapacitance strategy.

these redox peaks. At the remaining potential range, the increase of $b$-value indicates the characteristic of the pseudocapacitive effect (Supplementary Eq. 8). The pictorial estimation represented in Fig. 5e, defined that pseudocapacitance accounts for 34.4% of the total capacity at the scan rate of 0.1 mV s$^{-1}$.

Based on a similar procedure, we conducted ABR synthesis of various compounds and applied them as active materials for LIBs and SIBs. A brief summary on the electrochemical performance of these ABR compound is included in Supplementary Table 15; these values strongly confirm the prospective application of ABR strategy for the synthesis of active materials for low-cost and sustainable energy storage device in the near future.

## Discussion
Herein we report a green and environmentally friendly pathway of humidity-assisted ABR for the synthesis of a variety of active materials for LIBs and SIBs. A combination of ABR and Ca-

## Methods
**Chemical**. LiOH H$_2$O (98%) was purchased from JUNSEI Chemical. Vanadium (V) oxide, V$_2$O$_5$ (99.96%) and calcium oxide, CaO (99.9%) were obtained from Sigma-Aldrich.

**Preparation pure and Ca-doped samples**. Li$_3$VO$_4$ and deferent Ca-doping content samples were prepared via humid-assisted solid-state route based on ABR directly from precursors of LiOH, V$_2$O$_5$, and CaO. In a typical process, LiOH H$_2$O, CaO, and V$_2$O$_5$ with the stoichiometric mole percentage ($x$% = 0, 1, 3, and 5%) was well mixed and ground in mortar. The obtained mixture was transferred into a 20 mL glass vial which was added another open 3 mL vial containing 1 mL of distilled water. Then, the whole container was maintained in oven at 80 °C. After 24 h reaction, the water-vial was taken out, and the samples were dried at the same temperature for another 12 h before being re-ground and denoted as $x$LCVO-ABR.

For comparison, pure and 3% Ca-doped Li$_3$VO$_4$ was synthesized via SSR following reference[9]. After annealing process, as-prepared samples were re-ground and denoted as 0LCVO-SSR and 3LCVO-SSR, respectively.

**Structural and physicochemical characterization.** The structural information of as-synthesized samples was obtained from XRD on Philips X'Pert (Cu Kα radiation) in the 2θ range from 10 to 80° with a step size of 0.026°. The Rietveld refinement was resolved on the general structure analysis system program[65]. The particle size and morphology of samples were determined by FESEM (S-4700, Hitachi). The element distribution images were collected from energy-dispersive X-ray spectroscopy mapping methods attached on FESEM. The analysis on the valence state of elements was investigated by XPS measurement carried on VG MultiLab 2000, Thermo Scientific. The specific surface area of particles obtained by the BET method and the pore size distribution recorded by Barrett–Joyner–Halenda were carried out on BELSORP-mini II, BEL Japan, Inc. The electron paramagnetic resonance (EPR) spectra were acquired on a JEOL JES FA200 EPR spectrometer at room temperature.

**Electrochemical characterization.** The anode electrodes were prepared via the slurry method with mass ratios of 75:20:5 of active materials, conductive agent (Super-P), and binder poly (vinylidene fluoride) in N-Methyl-2-pyrolidone as solvent. The slurry was well mixed in mortal for 30 min then coated on the copper foil as a current collector, then the electrode was dried at 80 °C in vacuum state overnight before cutting into a disc with 14 mm diameter by a punch. The estimated total active material loaded on a single electrode disc is around 1 mg. The half cell using lithium metal foil as a counter electrode, polymer membrane as a separator, was assembled in Ar-filled glovebox according to the 2032 coin-cell model. The 1:1 mixture of ethylene carbonate and dimethyl carbonate dissolved 1 M LiPF$_6$ was used as electrolyte. The fabricated cell was aged for at least 6 h prior to electrochemical measurements.

Charge–discharge profiles were measured on a NAGANO BTS-2004H battery charger in the voltage range of 0.2 –3.0 V vs. Li/Li$^+$ at a current density of 100 mAh g$^{-1}$. The cycling voltammetry (CV) surveys were conducted on Autolab electrochemical workstation between voltage of 0.2 and 3.0 V vs. Li/Li$^+$. The EIS tests were performed using the same Autolab with a voltage of 5 mV amplitude in a frequency range of 100 kHz to 0.01 Hz.

The full cells were constructed using LiNi$_{0.6}$Mn$_{0.2}$Co$_{0.2}$O$_2$ cathode and 3LCVO-ABR anode or graphite, in which the active material mass loading on electrode was adjusted toward an anode/cathode capacity ratio of 0.9. All obtained full cells were cycled in voltage window of 2.5–4.5 V and the relevant specific capacities were calculated based on the mass of active material of anode.

**DFT calculation method.** To construct a Ca-doped Li$_{3-x}$Ca$_x$VO$_4$ model, we identified the more energetically favored Ca doping location among two irreducible Li sites (Li1 and Li2) in a Li$_3$VO$_4$ unit cell. One Ca dopant was located at the energetically favored doping site, Li2, inside an extended 2 × 2 × 2 Li$_3$VO$_4$ supercell (Li$_{48}$V$_{16}$O$_{64}$), which applied for all subsequent DFT calculations. The second Ca dopant was placed on another Li2 site to set the Ca atomic ratio (2/48 = 4.17 at.%) close to the experimental value (3.0 %). The location of the second Ca dopant was identified by calculating the Ca substitution energy of all available Li2 sites in the presence of the first Ca dopant. The final Li$_{46}$Ca$_2$V$_{16}$O$_{64}$ supercell (Supplementary Fig. 8a, b) was used for electronic analysis.

We performed hybrid-level spin-polarized DFT calculations using a plane-wave basis with the VASP code[66] and the HSE06 functional[67]. The ratio of the exact Hartree–Fock exchange was kept to 25%, a genuine value of HSE06. The initial Li$_3$VO$_4$ unit cell structure was optimized using the PBE functional. However, all subsequent calculations including geometry optimization was performed using the HSE06 functional. To appropriately treat the V d-orbitals, the Hubbard U formalism, DFT + U[68], with $U_{eff}$ = 3 eV[69,70] was applied for V ions. The projector augmented wave method was applied to describe the interaction between the ionic core and the valence electrons[71]. Valance electron wave functions were expanded in a plane-wave basis up to an energy cutoff of 500 eV. The Brillouin zone was sampled at the Γ-point for geometry optimization and a 2 × 2 × 2 k-points grid sampling was applied for electronic structure analysis. The convergence criteria for the electronic structure and the atomic geometry were set to 10$^{-4}$ eV and 0.03 eV/Å, respectively. We used a Gaussian smearing method with a finite temperature width of 0.05 eV to improve convergence of states near the Fermi level. The location and energy of transition states were calculated with the climbing-image nudged-elastic-band method with 9 or 15 images[72].

## Data availability

The data that support the findings within this paper are available from the corresponding author on request.

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

## Acknowledgements

This research was supported by the Basic Science Research Program through the National Research Foundation of Korea (NRF), funded by the Ministry of Science, ICT and Future Planning (NRF-2017R1A2B3011967). This work was supported by the Engineering Research Center through the NRF, funded by the Korean Government (MSIT), (NRF-2018R1A5A1025224). This research was supported by BK21 Four program in Hanyang University.

## Author contributions

H.T.H. conceptualization, synthesizing materials, conducting physico- and electrochemical analysis, writing manuscript; N.H.V. conducting electrochemical analysis; H.W.H. and H.Y.K. conducting DFT calculation; J.H.M. conducting in situ Raman spectroscopy; W.B.I. conceptualization, correcting manuscript and administering projects.

## Competing interests

The authors declare no competing interests.
