## [Peer Review File · Nature Communications]

Reviewer #1 (Remarks to the Author):

In this manuscript, authors proposed a relatively new synthesis strategy for the fabrication of Li₃VO₄ anode with Ca doping, showing a facile green chemistry process and tailored particle size and pore size distribution. The corresponding materials achieved good performance compared with pristine samples and previous reported studies on the same material, which were comprehensively elucidated by varied characterizations and DFT simulations. Therefore, the referee recommend it to publish in Nature Communications after a major revision. Following issues need to be address:

1. More discussion about the pre-intercalated of Ca should be added and compared with transition/non-metal doping system to show the advantages of the design.
2. The authors mentioned that the induced dopant (Ca) can have impact on lattice parameters of as-prepared samples by replacing Li from the accommodated sites and further identified the resulting particle sizes, it is also necessary to present crystal size corresponding to identified phases in demonstrated samples, since long/short-range order and grain boundary are also affect on the features of solid ionic diffusions.
3. There are some wrong statement of Raman analysis such as claimed relationships of Raman shifts at low frequency modes with V-O or V=O bond. Thus, it should be more clarified in Raman results with detailed elucidation of reaction evolutions and cited literature.
4. It should be given a detailed elemental analysis of EDS mapping, specifically on individual ABR related particle from high-resolution TEM, otherwise, it is hard to distinguish the homogeneity of elemental distribution in as-obtained samples via the proposed reaction approach. Moreover, the veritable concentration of dopant in treated sample should be addressed by using ICP-MS or XRF technology.
5. The analysis of XPS V core level spectra is not fitted correctly as the visually identified areal relationship (2p 3/2 and 2p 1/2) of related orbits in Vanadium are not matched with each other, which can give rise to significant misinterpretation of chemical states. Hence, authors should check it carefully. The peak fitting should be carefully checked, some of the peaks have too broad FWHM.
6. The diffusion energy barrier and the diffusion path of Li⁺ calculated by DFT should given so as to bring further insight into the diffusion process in accordance with observed enhanced diffusion kinetics, since the dopant occupied the indigenous Li sites and defective oxygen occurred in the lattice.
7. The authors mentioned "The higher CE of 3LCVO samples compared to the pure samples demonstrates the positive effect of Ca doping in enhancing the structural flexibility under deeply lithium exchange." therefore, it should be also given crystal structural evolutions during charge/discharge process to verify the claimed reversibility. More structural information related to the initial change should be uncovered.
8. In addition to Coulombic efficiency during GCD cycling tests in Figure 4b, the right Y axis should be re-scaled so that readers can see more clearly the fluctuation of the Coulombic efficiency.
9. The combined ohmic resistance fitted from EIS characterizations should be given.
10. Low frequency region should be enlarged for EIS.
11. The authors are suggested to compare the pre-lithiated anode and with the non-lithiated one. Both reference 58 and 59 are related to anodes without Li in the initial structure, but in this materials, the Li is in the crystal structures and with a rich content.
12. The materials show a good performance, a performance comparison with other anodes, especially with the state of the art Li₃VO₄ materials should be added in a form.
13. There are quite a few typos in the paper that the authors need to revised carefully. Such as Page 7, line 143, "the precursor particles is", etc.

Reviewer #2 (Remarks to the Author):

In this work, the authors created a new green-chemistry strategy with the addition of water vapor to fabricate high-performance Li_3VO_4 -based anode material for lithium-ion batteries. Based the delicate in-suit Raman characterization, the authors reasonable explain the mechanism of the role of the water droplets in the material preparation process. However, the electrochemical characterization for the materials is quite insufficient.

1. In the Experimental part, the author declared "The estimated total active material loaded on a single electrode disc is around 10 mg". The author should carefully check whether the mass of active material is wrong or the mass of the current collector was included. The loading mass of active material in the laboratory is usually difficult to reach such a large level.

2. The EIS analysis appears an obvious error: the manuscript declared that the calculated Li^+ ion diffusion coefficient of the optimal 3LCVO-ABR sample is the lowest among the series samples. This is not consistent the data listed Table 4. Why does the optimal 3LCVO-ABR sample fail to show the largest Li^+ diffusion coefficient, but the highest capacity? The author should provide reasonable explanations for the contradictory result.

3. The low initial CE of Li_3VO_4 was ascribed to the crystallite distortion due to lithium ions insertion. In the Ca-doped samples, more lithium ions could insert into the material that should induce a more distorted lattice. However, the CE of 3LCVO samples is higher than that of the pure sample. The authors just explain it as the doped samples can present a structural flexibility, this is not enough.

4. With increasing Ca-doping content, the concentration of oxygen vacancies V_o obviously increased. If the enhanced performance are attributed to the increased V_o concentration instead of just the higher surface area? Or, both of them are contributed to the highest capacity of the 3LCVO-ABR sample. Moreover, $4.484 \text{ m}^2 \text{ g}^{-1}$ is not a larger value of surface area for anode material. If such a surface area value can bring so larger capacitance increase of 3LCVO-ABR sample compared to the undoped material?

5. In TEM analysis part, the manuscript declared "While the edge surface of 0LCVO-ABR (Figure S4d) was tightly constructed with very less pores, the surface of 3LCVO-ABR (Figure 2c) clearly exhibited a high porosity with a loose stacking of nanoparticles". Why the Ca-doping sample 3LCVO-ABR has more pores than the pristine 0LCVO-ABR?

6. The discharge and charge profiles and CV curves are inconsistency, especially for the first discharge curve. The discharge and charge profiles declared "A typical discharge profile includes a sharp slope at voltages higher than 0.8 V attributed to lithium ion insertion into the lattice and a smooth region with two characteristic plateaus at ~ 0.75 and 0.6 V indicative of the interaction between active materials and the electrolyte, leading to the formation of a solid electrolyte interphase (SEI) layer in the first few cycles", while the CV curves demonstrated "This means that based on the doping strategy, peaks corresponding to reduction tend to shift to higher voltage regions while the oxidation peak moves toward lower potentials which reduces the voltage gap from 0.668/0.322 V for pristine samples to 0.504/0.212 V in 3LCVO-ABR". The above descriptions show two different electrochemical reaction mechanism, insertion or conversion type? The authors should provide more reasonable explanations and supply more characterizations for verifying the electrochemical reaction mechanism.

7. The cycling performance of Li_3VO_4 anode at much higher rate (10C) and longer cycles (>1000) was frequently reported previously. However, the cycling performance in present work is only characterized at a much lower rate of 100mA/g and limited cycles of 200. What the high-rate and long-cycle cycling performance of the studies samples.

8. The unit “mAh/g” has been written as “mA/g” in several places.

Reviewer #3 (Remarks to the Author):

In this manuscript, authors developed acid-base reactions to fabricate Li_3VO_4 with controllable morphology and particles size. In addition, a green combination of the acid-base reactions strategy and Ca doping was employed to enhance the electrochemical properties of Li_3VO_4 . This work is significant in green synthesis of Li_3VO_4 anode, and optimization mechanism of Li_3VO_4 . However, the Ca doping Li_3VO_4 in this work is less competitive compared with other Li anode materials. Therefore, I can't recommend this paper to publish in Nature Communications. The following are some comments:

1. In recent years, various novel lithium ion battery anode materials have been extensively reported. Many of them have been close to practical applications. According to the overall performance of Ca doping Li_3VO_4 in this work, it's still far away from realizing practical applications.
2. The TEM mappings of 3LCVO-ABR should be provided.
3. Although the synthesis method is green and facile in this work, it is less competitive compared with others methods such as solvent-free synthesis of silicon-based anode materials.
4. The Li_3VO_4 with different doping Ca contents possess different morphologies and structures, which may also affect the electrochemical performance.
5. A specific capacity of 543.1 mA h g⁻¹ at 100 mA g⁻¹ is not superior enough as many works have reported the same performance of Li_3VO_4 anode.
6. Decreasing the size of Li_3VO_4 particles is a regular method to improving its lithium storage performance, so it's hard to find the highlight of this article.

Reviewer 1

Overall comment: In this manuscript, authors proposed a relatively new synthesis strategy for the fabrication of Li_3VO_4 anode with Ca doping, showing a facile green chemistry process and tailored particle size and pore size distribution. The corresponding materials achieved good performance compared with pristine samples and previous reported studies on the same material, which were comprehensively elucidated by varied characterizations and DFT simulations. Therefore, the referee recommend it to publish in Nature Communications after a major revision. Following issues need to be address.

We appreciate the Reviewer's valuable comments on our manuscript and the additional suggestions We have revised the manuscript accordingly.

Comment 1: More discussion about the pre-intercalated of Ca should be added and compared with transition/non-metal doping system to show the advantages of the design.

Response: The authors thank the Reviewer's suggestion. In the revised version, we have added further discussion on the advantages of Ca-doping compared to non-/transition metal doping strategies as shown below.

Compared to morphology engineering or composite fabrication, aliovalent substitution strategy was demonstrated as a direct and effective way to modify the electronic structure leading to enhanced electronic conductivity. Y. Dong et al. has report that Mo^{6+} doping to V^{5+} could alter electronic band structure of Li_3VO_4 as n-type semiconductor and shift Fermi level toward conduction band due to induced extra electrons. Meanwhile, Ni-doped Li_3VO_4 with an improved surface energy could

accelerate the insertion/extraction of Li^+ ions. Besides, substitution of Li^+ by Na^+ could enhance electrochemical performance of Li_3VO_4 due to lattice parameter enlargement and particles size reduction. Furthermore, the Mg^{2+} introduction to Li^+ sites not only lead to lattice expansion but also enhance electronic conductivity leading to improvement in electrochemical properties. Therefore, Ca^{2+} which is same group of Mg^{2+} with larger ionic radius could be a good candidate for doping to Li_3VO_4 .

(Revised manuscript, page 4)

Comment 2: The authors mentioned that the induced dopant (Ca) can have impact on lattice parameters of as-prepared samples by replacing Li from the accommodated sites and further identified the resulting particle sizes, it is also necessary to present crystal size corresponding to identified phases in demonstrated samples, since long/short-range order and grain boundary are also affect on the features of solid ionic diffusions.

Response: We thank Reviewer for the comment. As recommendation of Reviewer, we determined the crystallite size and microstrain using Williamson – Hall plot. The linear plots of $\beta\cos\theta$ vs. $4\sin\theta$ of pure and doped samples from ABR and SSR method were present in Figure S21 and the obtained crystallite size and microstrain were summarized in Table S7. Accordingly, the doped samples perform lower slopes and higher y-intercept values which indicates a minimization of lattice strain and crystallite size after Ca-doping. These observation could partly attribute to the enhancement in electrochemical performance of Ca-doped and ABR samples. The reduction in crystallite size could increase the grain boundary and lead to more lithium ion take part in the inter-/deintercalation process. Meanwhile, the microstrain is well-known related to the crystal and then particles cracking formation which cause the capacity fading. Therefore, minimizing the

microstrain can reduce the lattice mismatching and improve materials' cyclability. In revised manuscript, we have added this calculation and discussion as shown below.

Additionally, the enhancement in lithium ion diffusion of doped samples is also originated from the crystallite size calculated by Williamson – Hall (WH) plot. As shown in Figure S21 and summarized in Table S7, there is a significant decrease in crystallite size of 3LCVO-ABR compared to the remaining samples leading to increase of grain boundary at which ionic conductivity is much higher than grain interior. Therefore, a reduction in crystallite size along with particle size could accelerate the enhanced ionic conductivity of doped sample. Furthermore, the microstrain, which originates the micro-crack and pulverization of particles, extracted from WH plot (Table S7) exhibits a significant decrease in 3LCVO-ABR compared to pure samples. This less lattice mismatching observed after Ca-doping is favorable for improving cycling performance of active materials.

(Revised manuscript, page 16)

Figure S21. Williamson–Hall (WH) plot of the xLCVO-ABR and -SSR samples based on the XRD pattern showing the lattice strain and crystallite size variation. WH equation: $\beta \cos\theta = K\lambda/D + 4\epsilon \sin\theta$, in which β is full width at half maximum of the diffraction peak (rad); θ is Bragg angle (rad); K is shape factor (Scherrer constant approximately 1), λ is wavelength of X-ray source (1.5418 \AA); D is crystallite size (\AA); and ϵ is microstrain.

(Revised Supporting Information, page S31)

Table S7. Crystallite size and microstrain of xLCVO-ABR and –SSR calculated by Williamson – Hall plots.

samples	Williamson – Hall fitting equation	crystallite size, D (Å)	lattice microstrain, ϵ
0LCVO-ABR	$y = 0.00182x + 0.00407$	378.82	0.00182
3LCVO-ABR	$y = 0.00034x + 0.00859$	179.49	0.00034
0LCVO-SSR	$y = 0.00309x + 0.00019$	8114.74	0.00309
3LCVO-SSR	$y = 0.00273x + 0.00069$	2234.49	0.00273

(Revised Supporting Information, page S42)

Comment 3: There are some wrong statement of Raman analysis such as claimed relationships of Raman shifts at low frequency modes with V-O or V=O bond. Thus, it should be more clarified in Raman results with detailed elucidation of reaction evolutions and cited literature.

Response: We appreciate for reviewer’s recommendation. In revised manuscript, we have carefully discussed again about our *in situ* Raman data as shown below.

At the beginning of the reaction, LiOH was characterized by an intense peak located at 1091.6 cm^{-1} in the spectrum generated at position 2. Meanwhile, the highest intensity Raman peak of position 1 centered at 991.3 cm^{-1} could be assigned to stretching mode (A_{1g} and B_{2g}) of vanadyl group V=O. A low intensity signal located at 951.3 cm^{-1} is attributed to the asymmetric stretching (B_{2g}) vibration of the bridging bond V-O₍₃₎-V while the “rail” of the ladder of bridging V-O₍₂₎-V bonds with out-of-phase stretching vibration (B_{1g} and B_{3g}) could be assigned for peak at 702.5 cm^{-1} . The signal at 528.3 cm^{-1} could be ascribed to the stretching mode (A_{1g}) of V-O₍₂₎ “ladder steps”. The V-

O₍₃₎-V bridge with symmetric stretching vibration (A_{1g}) could be observed at Raman shift of 482.7 cm⁻¹ while peak at 403.2 cm⁻¹ could be accounted for the bending mode (A_{1g}) of this bridging bonds. The contiguous peaks at 292.1 and 281.4 cm⁻¹ could be assigned to the ladder-like distortion which is a combination of the in-phase vibration of V and O₍₂₎ atoms and the out-of-phase oscillation of O₍₃₎ atoms along the *b*-axis. The ladder breathing mode and the V=O₍₁₎ swinging vibration could be attributed to the signal at 201.1 cm⁻¹ while the lowest Raman shift signal at 147.6 cm⁻¹ (B_{1g}) could be accounted for the shearing-like distortion which is originated from the symmetric vibration of V, O₍₂₎, and O₍₃₎ atoms along the *b*-direction.¹ After 90 min of reaction, the peaks' intensity at position 1 decreased and this effect might be attributed to the surrounding of water layer as well as dissolving consumption by droplet. Whereas, the spectrum of position 2, beside peaks related to LiOH, new peaks in the range of 250 – 500 and 750 – 1000 cm⁻¹ corresponds to the formation of Li₃VO₄, indicating the higher reaction rate of these areas which can be ascribed to the high solubility of LiOH.^{2,3,4} Similar signals could be observed in the spectrum at position 3, which is somewhere with well-mixed precursors particles, demonstrating that the formation of Li₃VO₄ could occur at any delocalized position due to the mobility of the droplet-reactor. Additionally, the reappearance of peak at 1091 cm⁻¹ in Raman spectrum of position 3 after 360 min could prove the mobility of droplet reactors containing LiOH precursor. The reduction of LiOH-related peaks at ~ 1091 cm⁻¹ and the main contribution of Li₃VO₄ signal after 270 min of reaction could be demonstrated for the effect of mobile vapor droplets on difference of reaction rate. The low solubility of V₂O₅ in water inhibits the synthesis reaction on site of this precursor, which is confirmed by the

spectrum at position 1, in which only the peaks of V-precursor were remaining. In contrast, in other places, the formation of the final product was almost complete after 270 min.

(Revised manuscript, page 7, 8)

Comment 4: It should be given a detailed elemental analysis of EDS mapping, specifically on individual ABR related particle from high-resolution TEM, otherwise, it is hard to distinguish the homogeneity of elemental distribution in as-obtained samples via the proposed reaction approach. Moreover, the veritable concentration of dopant in treated sample should be addressed by using ICP-MS or XRF technology.

Response: We thank for Reviewer's comment. In revised version, we have added a HR-TEM mapping result as shown in Figure S5. The content of component elements was also analyzed using ICP-OES and XRF techniques and summarized in Table S5.

Figure S5. TEM elemental mapping of 3LCVO: (a) TEM image of mapping area; mapping elements as (b) V; (c) O; and (d) Ca.

(Revised Supporting Information, page S15)

Table S5. The elemental composition of xLCVO obtained from ICP-OES and XRF analysis.

	mass %Li (%)			mass %Ca (%)			mass %V (%)			mass %O (%)		
	TC	ICP-OES	XRF	TC	ICP-OES	XRF	TC	ICP-OES	XRF	TC	ICP-OES	XRF
0LCVO-ABR	15.44	14.96	-	0	0	0	37.50	37.91	36.14	47.06	-	48.15
1LCVO-ABR	15.31	14.65	-	0.29	0.18	0.31	37.43	36.01	37.12	46.96	-	47.01
3LCVO-ABR	15.05	13.14	-	0.88	0.76	0.93	37.27	35.02	36.21	46.79	-	46.02
5LCVO-ABR	14.78	13.65	-	1.46	1.12	1.36	37.15	37.06	36.94	46.61	-	45.21
0LCVO-SSR	15.44	13.02	-	0	0	0	37.50	37.81	36.54	47.06	-	48.42
3LCVO-SSR	15.05	12.42	-	0.29	0.25	0.30	37.43	37.21	37.51	46.96	-	44.12

(Revised Supporting Information, page S35)

Comment 5: The analysis of XPS V core level spectra is not fitted correctly as the visually identified areal relationship (2p 3/2 and 2p 1/2) of related orbits in Vanadium are not matched with each other, which can give rise to significant misinterpretation of chemical states. Hence, authors should check it carefully. The peak fitting should be carefully checked, some of the peaks have too broad FWHM.

Response: The authors appreciate to the Reviewer's suggestion. As Reviewer's request, we have conducted the XPS fitting again as shown below. We hope our new results could satisfy Reviewer's requirement. In detail, the FWHM and splitting energy of V2p3/2 – V2p1/2 of our fitting were summarized in the Table R1. Accordingly, the splitting energy and FWHM of XPS peaks after fitting are close to the reference (splitting energy of 7.35 eV; FWHM of 0.94 eV for V⁵⁺2p3/2; 2.47 eV for V⁵⁺2p3/2; and 1.37 eV for V⁴⁺).[*Physical Review B*, **12**(1) (1975) 12-19]

(Revised Supporting Information, page S21)

Table R1. The summary of peak position, FWHM, and splitting energies derived from XPS fitting.

samples	V^{5+} (eV)			V^{4+} (eV)		
	V2p3/2 - FWHM	V2p1/2 - FWHM	splitting energy	V2p3/2 - FWHM	V2p1/2 - FWHM	splitting energy
0LCVO- ABR	517.42 – 1.08	525.05 – 2.32	7.63	516.35 – 1.42	523.93 – 1.46	7.58
3LCVO- ABR	517.54 – 1.02	525.02 – 2.41	7.48	516.77 – 1.38	524.37 – 1.40	7.6
3LCVO- ABR	517.62 – 1.13	525.12 – 2.45	7.50	516.56 – 1.40	523.88 – 1.42	7.32

Comment 6: The diffusion energy barrier and the diffusion path of Li⁺ calculated by DFT should be given so as to bring further insight into the diffusion process in accordance with observed enhanced diffusion kinetics, since the dopant occupied the indigenous Li sites and defective oxygen occurred in the lattice.

Response: We thank Reviewer for a constructive comment. As suggested by the reviewer, we estimated the diffusion pathway of a Li⁺ ion moving across the adjacent sites and calculated the corresponding diffusion energy barrier, E_{act} , using two models, $\text{Li}_{48}\text{V}_{16}\text{O}_{64}$ and $\text{Li}_{46}\text{Ca}_2\text{V}_{16}\text{O}_{63}$.

The DFT-calculated E_{act} values from two separated diffusion pathways (Li^{2a} to Li^{2a} and Li^{2a} to Li^{4b}) of a Li ion inside a $\text{Li}_{48}\text{V}_{16}\text{O}_{64}$ model show that the highest E_{act} along the multi-step diffusion pathway is as low as 0.36 eV (Li^{2a} to Li^{2a} , parallel) or 0.25 eV (Li^{2a} to Li^{4b} , perpendicular), confirming that quick Li (de)intercalation is available in both directions. On the other hand, the overall morphology of the diffusion pathways presented in Fig. R1b and R2b suggest that a Li⁺ ion diffuses inside a $\text{Li}_{46}\text{Ca}_2\text{V}_{16}\text{O}_{63}$ model stepwise due to the presence of the intermediate sites on the diffusion pathways, which are extra-stabilized by the Ca dopant and the oxygen vacancy. The highest E_{act} calculated in a $\text{Li}_{46}\text{Ca}_2\text{V}_{16}\text{O}_{63}$ model was increased (0.42 eV for parallel diffusion and 0.75 eV for perpendicular diffusion) from those calculated in a $\text{Li}_{48}\text{V}_{16}\text{O}_{64}$ model, suggesting that Li diffusion becomes sluggish upon Ca doping. However, interestingly, Fig. R2a and R2b comparatively show that the Li^{4b} site becomes energetically unstable upon Ca doping and therefore, the perpendicular Li^{2b} to Li^{4b} diffusion pathway is deactivated in $\text{Li}_{46}\text{Ca}_2\text{V}_{16}\text{O}_{63}$ and thus the (de)intercalation pathway of LCVO is highly regulated into single parallel type (Li^{2a} to Li^{2a}). Because the difference in the E_{act} values of Li diffusion presented in Fig. R1a and R1b is as low as 0.07 eV, the experimentally observed increased Li diffusivity in LCVO is originated presumably from the pathway regulation effect upon Ca doping. We are going to study intensively the diffusion

behavior of Li^+ ions inside LVO and LCVO. The result will be reported in due course. Furthermore, we also consider other factor which contributes to improve the ionic conductivity of Ca-doped samples such as crystallite size. As determined using Williamson – Hall plot, the doped sample perform a significant decrease in crystallite size which could induced higher mobility at grain boundary. Therefore, beside the regulation to one diffusion pathway, Ca-doping could enhance ionic conductivity of Li_3VO_4 by reducing crystallite size.

In this revision, we also conducted the calculation on electronic band diagram to demonstrate effect of Ca-doping on reduction of bandgap which could induce the improvement in electronic conductivity. As we have discussed, the main effect of dopant to Li-site is attributed to the modify the electronic configuration and reduce bandgap leading to improve the electronic conductivity. According to our calculation, we propose band diagram of pure sample (0LCVO), Ca-doped Li_3VO_4 without (3LCVO) and within oxygen vacancy (3LCVO-1VAC), as shown in Figure R3. The doped sample performs a significant reduction in gap between valence band and conduction band. Furthermore, the occupation of electrons at intermediate level which are attributed to the formation of excess electrons induced from Ca-doping are more freely to excite to conduction band. These results demonstrated the origin of effect on enhanced electronic conductivity of Ca-doped sample.

We also modified our discussion in revised manuscript as below.

In order to clarified the effect of Ca-doping on electronic and ionic conductivity of Li_3VO_4 , a theoretical calculation has been conducted. Herein, we have conducted calculation on the energy barrier of lithium ion diffusion in two pathway: *i*) parallel $\text{Li}^{2a} - \text{Li}^{2a}$ transport, and *ii*) perpendicular $\text{Li}^{2a} - \text{Li}^{4b}$ hoping. Accordingly, the highest activation energies (E_{act}) of lithium ion diffusion in $\text{Li}_{48}\text{V}_{18}\text{O}_{64}$ supercell are 0.36 eV

($\text{Li}^{2a} - \text{Li}^{2a}$ parallel, Figure S14a) and 0.25 eV ($\text{Li}^{2a} - \text{Li}^{4b}$ perpendicular, Figure S15a) which demonstrates that both directions are favorable for transport of lithium ions. However, the overall morphology of lithium ion diffusion in model of $\text{Li}_{46}\text{Ca}_2\text{V}_{18}\text{O}_{63}$ is a stepwise process within intermediate position stabilized by the presence of Ca-substitution and oxygen vacancy. The increase of highest E_{act} in $\text{Li}_{46}\text{Ca}_2\text{V}_{18}\text{O}_{63}$ model determined the sluggish effect of Ca-doping on diffusion of lithium ions. However, the significant increase in energy barrier as shown in Figure S15a,b indicates the deactivation of perpendicular pathway in doped sample while the slight raise of 0.06 eV in parallel movement is acceptable.⁵ Therefore, a regulation of lithium ion diffusion in Ca-doping case into single parallel type could be presumable origin for the enhancement in ionic conductivity observed in experimental data. Additionally, the enhancement in lithium ion diffusion of doped samples is also originated from the crystallite size calculated by Williamson – Hall (WH) plot. As shown in Figure S16 and summarized in Table S6, there is a significant decrease in crystallite size of 3LCVO-ABR compared to the remaining samples leading to increase of grain boundary at which ionic conductivity is much higher than grain interior.⁶ Therefore, a reduction in crystallite size along with particle size could accelerate the enhanced ionic conductivity of doped sample. Furthermore, the microstrain, which originates the micro-crack and pulverization of particle, extracted from WH plot (Table S6) exhibits a significant decrease in 3LCVO-ABR compared to pure samples. This less lattice mismatching observed after Ca-doping is favorable for improving cycling performance of active materials.⁷ In addition, the electronic band diagrams were present in Figure S17 and the information of electron occupation was summarized in

Table S7-12. According to Figure S17, the valence band maximum (VBM) and conduction band minimum (CBM) are -0.7016 eV and 5.1450 eV, corresponding to a bandgap of 5.8466 eV. The introduction of Ca^{2+} into lattice of $\text{Li}_{46}\text{Ca}_2\text{V}_{18}\text{O}_{64}$ induced a significant shift of both VBM and CBM toward a lower band gap of 4.5061 eV. In addition, the excess electrons induced from charge compensation could occupy the energies level higher than VBM which means more probability for electron to excite to conduction band. The similar result was also observed in additional formation of oxygen vacancy. Furthermore, the reduction effect of these excess electrons could induce the formation of mixed oxidation state of $\text{V}^{4+}/\text{V}^{5+}$ which shift the Fermi level toward conduction band. Therefore, beside the enhancement in ionic diffusion, the electronic conductivity improvement of doped samples could be attributed to the modification of Ca-doping on bandgap structure.

(Revised manuscript, page 16, 17)

Figure R1. DFT-estimated parallel diffusion pathway of Li ion from the Li^{2a} to the adjacent Li^{2a} site in (a) a $\text{Li}_{48}\text{V}_{16}\text{O}_{64}$ or (b) a $\text{Li}_{46}\text{Ca}_2\text{V}_{16}\text{O}_{63}$ model. The migrating Li ion is colored in pink. The

black colored numbers present the relative energy of the corresponding state. The E_{act} denotes the diffusion energy barrier of each step.

Figure R2. DFT-estimated perpendicular diffusion pathway of Li ion from the Li^{2a} to the adjacent Li^{4b} site in (a) a $Li_{48}V_{16}O_{64}$ or (b) a $Li_{46}Ca_2V_{16}O_{63}$ model. The migrating Li ion is colored in pink. The black colored numbers present the relative energy of the corresponding state. The E_{act} denotes the diffusion energy barrier of each step.

Figure R3. Band diagram derived from DFT calculation for pure Li_3VO_4 (0LCVO); Ca-doped Li_3VO_4 (3LCVO); and Ca-doped Li_3VO_4 with oxygen vacancy (3LCVO-1VAC).

Comment 7: The authors mentioned “The higher CE of 3LCVO samples compared to the pure samples demonstrates the positive effect of Ca doping in enhancing the structural flexibility under deeply lithium exchange.” therefore, it should be also given crystal structural evolutions during charge/discharge process to verify the claimed reversibility. More structural information related to the initial change should be uncovered.

Response: We appreciate Reviewer’s comment. According to the DFT calculation on diffusion properties of doped sample, it is observable that Ca-doping along with formation of oxygen vacancy raise a stabilization of lithium ion to intercalation and transport. In addition, the reduction in microstrain of 3LCVO-ABR compared to pure sample could also be found. These effects could be considered as origin for better structural flexibility of doped sample. Furthermore, we have conducted the *in situ* XRD to clarify the structure evolution of electrode during the first charge and discharge process. As shown in Figure S14, the new phase formation could be indexed after discharged to 0.8 V in case of pure Li_3VO_4 while there is no secondary peak could be observed in pattern of 3LCVO-ABR. This illustrates that the structure of Li_3VO_4 after Ca-doping can adapt more lithium ion to insert/extract without phase transformation which is corresponding to the irreversible capacity loss in the first cycle. [*ChemElectroChem*, 2020, 7.9: 2033-2041.] Therefore, our discussion on effect of Ca-doping on enhancement of structural flexibility is demonstrated. We also provide additional discussion in revised manuscript as below.

According to the *in situ* XRD shown in Figure S14, while lithiation/delithiation in the first cycle of 0LCVO-ABR perform a formation of unknown peak (phase II) related to distortion of pristine structure to the secondary phase which is considered as main cause for the irreversible capacity loss in the first cycle, there is no new phase could

be observed in the XRD pattern of 3LCVO electrode. This result could illustrate for the effect of Ca-doping on regulating the lattice structure of Li_3VO_4 toward higher adaptability of lithium ions insertion/extraction.

(Revised manuscript, page 14)

Furthermore, the microstrain, which originates the micro-crack and pulverization of particle, extracted from WH plot (Table S6) exhibits a significant decrease in 3LCVO-ABR compared to pure samples. This less lattice mismatching observed after Ca-doping is favorable for improving cycling performance of active materials. Along with the stabilization of lithium ion after Ca-doping observed in DFT calculation, the reduction of microstrain is consistent to the previous discussion on enhancement in structural flexibility which originates the better first CE.

(Revised manuscript, page 17)

Figure S14. *In situ* XRD of (a) 0LCVO-ABR; and (b) 3LCVO-ABR for the first cycle of discharge/charge in 2θ -range of 20 - 35°.

(Revised Supporting Information, page S24)

Comment 8: In addition to Coulombic efficiency during GCD cycling tests in Figure 4b, the right Y axis should be re-scaled so that readers can see more clearly the fluctuation of the Coulombic efficiency.

Response: We thank Reviewer for recommendation. Herein, an enlargement of Coulombic efficiency at current density of 100 mA·g⁻¹ has been plotted in Figure S13.

Comment 9: The combined ohmic resistance fitted from EIS characterizations should be given.

Response: We appreciate Reviewer for suggestion. In our revised manuscript, we have added supplementary information of Ohmic resistance and charge transfer resistance as below.

Table 4. EIS Charge transfer resistances, linear relation of Z' versus $\omega^{-1/2}$, and lithium-ion diffusion coefficients of the samples.

samples	Ohmic resistance (Ω)	charge transfer resistance (Ω)	σ ($\Omega \cdot s^{-0.5}$)	D_{Li^+} ($cm^2 \cdot s^{-1}$)
0LCVO-SSR	4.22	87.81	69.206	6.4×10^{-11}
0LCVO-ABR	7.90	71.65	41.368	1.07×10^{-10}
1LCVO-ABR	4.35	62.34	101.397	4.4×10^{-11}
3LCVO-ABR	3.64	57.64	41.057	1.08×10^{-10}
5LCVO-ABR	2.85	67.69	46.729	9.4×10^{-11}
3LCVO-SSR	3.88	78.98	111.900	3.9×10^{-11}

Comment 10: Low frequency region should be enlarged for EIS.

Response: We appreciate Reviewer for suggestion. In our revised manuscript, we have added an enlargement of Nyquist plot of EIS data as shown in Figure S16.

Comment 11: The authors are suggested to compare the pre-lithiated anode and with the non-lithiated one. Both reference 58 and 59 are related to anodes without Li in the initial structure, but in this materials, the Li is in the crystal structures and with a rich content.

Response: We thank Reviewer for this question. We would like to give an explanation to help Reviewer to understand our experiment. Due to the common formation of SEI layer which cause the irreversible capacity loss in a few first cycles, the 3LCVO-ABR performs a capacity fading in around its 20 initial cycles in half cell. Therefore, in order to eliminate this capacity loss in full cell which cycles with limited lithium source, we have cycled the 3LCVO-ABR in half cell for 20 cycle until it reaches to stably cycling period. After that, the half cell was opened in Ar-filled glove box and used as anode for full cell assembly. We have wrong statement when using term of “pre-

lithiated” and we have change our text in revised manuscript as below. We hope that our explanation could clarify the misunderstanding of Reviewer.

To eliminate rapid capacity fading caused by irreversible lithium consumption in the first few cycles, the 3LCVO-ABR electrode was cycled for 20 cycles in another half-cell at a current density of $50 \text{ mA}\cdot\text{g}^{-1}$ before full-cell assembly.

(Revised manuscript, page 18)

Comment 12: The materials show a good performance, a performance comparison with other anodes, especially with the state of the art Li_3VO_4 materials should be added in a form.

Response: We thank Reviewer for recommendation. In our revised version, we have conducted a brief summary and comparison on electrochemical performance of our material and previously reported Li_3VO_4 -based anode as shown in Table S13.

Comment 13: There are quite a few typos in the paper that the authors need to revised carefully. Such as Page 7, line 143, "the precursor particles is", etc.

Response: We thank Reviewer for pointing out our mistakes. We have carefully checked up and corrected those typos in revised manuscript.

Reviewer 2

Overall comment: In this work, the authors created a new green-chemistry strategy with the addition of water vapor to fabricate high-performance Li_3VO_4 -based anode material for lithium-ion batteries. Based the delicate in-suit Raman characterization, the authors reasonable explain the mechanism of the role of the water droplets in the material preparation process. However, the electrochemical characterization for the materials is quite insufficient.

We appreciate the reviewer's valuable comments on our manuscript. We have revised the manuscript accordingly with additional experimental and theoretical calculation results. We hope our revision could reach Reviewer's requirement for further acceptance.

Comment 1: In the Experimental part, the author declared "The estimated total active material loaded on a single electrode disc is around 10 mg". The author should carefully check whether the mass of active material is wrong or the mass of the current collector was included. The loading mass of active material in the laboratory is usually difficult to reach such a large level.

Response: We thank Reviewer for comment. We admitted that it is our typing error. The active materials mass loading in a single electrode is 1 mg. We have corrected this fault in revised version.

Comment 2: The EIS analysis appears an obvious error: the manuscript declared that the calculated Li^+ ion diffusion coefficient of the optimal 3LCVO-ABR sample is the lowest among the series samples. This is not consistent the data listed Table 4. Why does the optimal 3LCVO-ABR sample fail to show the largest Li^+ diffusion coefficient, but the highest capacity? The author should provide reasonable explanations for the contradictory result.

Response: We appreciate Reviewer for pointing out our wrong statement. Actually, we would like to state that the lowest slope of linear plot between real impedance and angular frequency could derive to the lowest Warburg coefficient or the highest diffusion coefficient of 3LVO-ABR which indicates the highest ionic conductivity of this sample. We have correct our discussion as below. The lowest slope of linear plot indicating the highest Li^+ ion diffusion coefficient obtained for 3LCVO-ABR illustrates that the mobility of Li-ions was effectively accelerated by the expanding the lattice parameter and increasing the surface area of the electrode.

The lowest slope of linear plot indicating the highest Li^+ ion diffusion coefficient obtained for 3LCVO-ABR illustrates that the mobility of Li-ions was effectively accelerated by the expanding the lattice parameter and increasing the surface area of the electrode.

(Revised manuscript, page 16)

Comment 3: The low initial CE of Li_3VO_4 was ascribed to the crystallite distortion due to lithium ions insertion. In the Ca-doped samples, more lithium ions could insert into the material that should induce a more distorted lattice. However, the CE of 3LCVO samples is higher than that of the pure sample. The authors just explain it as the doped samples can present a structural flexibility, this is not enough.

Response: We appreciate Reviewer for comment. To demonstrate the conclusion on enhancement effect of Ca-doping on accommodation of structural flexibility to the lithium ion insertion/extraction, we have conducted the *in situ* XRD on the first cycle. As shown in Figure S15a, after discharge to 0.8 V, the raise of new peaks located at 2θ of 22.5° and 23.6° could be ascribed to formation of secondary phase which indicate the distortion of pristine Li_3VO_4 with orthorhombic phase to monoclinic or triclinic. This distortion along with formation of SEI layer is

considered as main cause of irreversible capacity loss in the first cycles. [ChemElectroChem 2020, 7, 2033 –2041]. Meanwhile, the *in situ* pattern of 3LCVO-ABR does not perform the same behavior which indicates that the insertion of lithium ion can not induced a new phase formation. To clarify this phenomenon, a DFT calculation on diffusion pathway of lithium ions in case of pure and doped Li_3VO_4 was conducted. As result shown in Figure S18, the substitution of Ca to Li site and formation of oxygen vacancy could stabilize the intermediate occupancy of inserted lithium ion which could demonstrate the enhancement in adaptability of structure after doping. Furthermore, we also carried out the calculation on the microstrain determination using Williamson – Hall plot. Accordingly, the reduction of crystallite size and microstrain of doped sample illustrates the improvement effect on accommodation ability of Ca-doping. We have modified our discussion on this part as below.

According to the *in situ* XRD shown in Figure S14, while lithiation/delithiation in the first cycle of 0LCVO-ABR perform a formation of unknown peak (phase II) related to distortion of pristine structure to the secondary phase which is considered as main cause for the irreversible capacity loss in the first cycle, there is no new phase could be observed in the XRD pattern of 3LCVO electrode. This result could illustrate for the effect of Ca-doping on regulating the lattice structure of Li_3VO_4 toward higher adaptability of lithium ions insertion/extraction.

(Revised manuscript, page 14)

Furthermore, the microstrain, which originates the micro-crack and pulverization of particle, extracted from WH plot (Table S6) exhibits a significant decrease in 3LCVO-ABR compared to pure samples. This less lattice mismatching observed after Ca-doping is favorable for improving cycling performance of active materials. Along with

the stabilization of lithium ion after Ca-doping observed in DFT calculation, the reduction of microstrain is consistent to the previous discussion on enhancement in structural flexibility which originates the better first CE.

(Revised manuscript, page 17)

Figure S18. DFT-estimated parallel diffusion pathway of Li ion from the Li^{2a} to the adjacent Li^{2a} site in (a) a $\text{Li}_{48}\text{V}_{16}\text{O}_{64}$ or (b) a $\text{Li}_{46}\text{Ca}_2\text{V}_{16}\text{O}_{63}$ model. The migrating Li ion is colored in pink. The black colored numbers present the relative energy of the corresponding state. The E_{act} denotes the diffusion energy barrier of each step.

Figure S19. DFT-estimated perpendicular diffusion pathway of Li ion from the Li^{2a} to the adjacent Li^{4b} site in (a) a $\text{Li}_{48}\text{V}_{16}\text{O}_{64}$ or (b) a $\text{Li}_{46}\text{Ca}_2\text{V}_{16}\text{O}_{63}$ model. The migrating Li ion is colored in pink. The black colored numbers present the relative energy of the corresponding state. The E_{act} denotes the diffusion energy barrier of each step.

Comment 4: With increasing Ca-doping content, the concentration of oxygen vacancies V_o obviously increased. If the enhanced performance are attributed to the increased V_o concentration instead of just the higher surface area? Or, both of them are contributed to the highest capacity of the 3LCVO-ABR sample. Moreover, $4.484 \text{ m}^2 \text{ g}^{-1}$ is not a larger value of surface area for anode material. If such a surface area value can bring so larger capacitance increase of 3LCVO-ABR sample compared to the undoped material?

Response: The authors have a thank to the Reviewers' comments. We agree to Reviewer that the surface area of 3LCVO-ABR is not large. Therefore, the contribution of capacitance raising from surface related process only accounted for 34.4% at scan rate of $0.1 \text{ mV}\cdot\text{s}^{-1}$. In addition, because this surface area is not enough to perform a large capacity as electrochemical double layer capacitance, a pseudo-capacitive process is more reasonable to explain for the excess capacity in

Ca-doped sample. In pseudo-capacitive process, there are three types of mechanism, including: *i*) underpotential deposition; *ii*) surface redox reaction; and *iii*) intercalation pseudo-capacitance. Therefore, beside surface area, other factor could affect to contribution of pseudo-capacitance such as surface defect, porosity, pore size distribution. For instance, as shown in O1s XPS (Figure S10), the surface of 3LCVO-ABR is consisted of large amount of oxygen defect which could be favorable for deposition of lithium ion.[Chem. Commun., 2017, 53, 12410-12413] In our work, we are not able to calculate how much contribution of oxygen vacancy on capacity. However, as result of DFT calculation, the stabilization effect of oxygen vacancy on lithium ion insertion/extraction could ascribe as the way how Ca-doping enhance the lithium storage in Li_3VO_4 . Beside that, the higher surface area of 3LCVO-ABR could be also consider as secondary contribution to the capacity enhancement.

Comment 5: In TEM analysis part, the manuscript declared “While the edge surface of 0LCVO-ABR (Figure S4d) was tightly constructed with very less pores, the surface of 3LCVO-ABR (Figure 2c) clearly exhibited a high porosity with a loose stacking of nanoparticles”. Why the Ca-doping sample 3LCVO-ABR has more pores than the pristine 0LCVO-ABR?

Response: Thank Reviewer for the question. As discussed in morphology part, we have propose a mechanism on how Ca-doping modifies the surface of Li_3VO_4 . Accordingly, the occupation of Ca^{2+} -ions, which possesses a lower hydration energies ($-1577 \text{ kJ mol}^{-1}$) compared to that of Li^+ -ions (-520 kJ mol^{-1}), on the surface could increase the lattice surface tension then leading to reduce the surface energy of growing particles. This stronger binding energy of Ca^{2+} to coordinated water molecules could prohibit the further reaction. In addition, compared to lithium ions, Ca-ions exhibit a lower diffusivity ($7.93 \times 10^{-10} \text{ m}^2 \cdot \text{s}^{-1}$ for Ca^{2+} -ion and $10.3 \times 10^{-10} \text{ m}^2 \cdot \text{s}^{-1}$ for Li^+ -ion).

Therefore, we assume that the synthesis reaction of Li_3VO_4 in Ca-ions presence is non-uniform crystallization leading to uneven surface. The above process was illustrated in Figure 2f.

Comment 6: The discharge and charge profiles and CV curves are inconsistent, especially for the first discharge curve. The discharge and charge profiles declared “A typical discharge profile includes a sharp slope at voltages higher than 0.8 V attributed to lithium ion insertion into the lattice and a smooth region with two characteristic plateaus at ~ 0.75 and 0.6 V indicative of the interaction between active materials and the electrolyte, leading to the formation of a solid electrolyte interphase (SEI) layer in the first few cycles”, while the CV curves demonstrated “This means that based on the doping strategy, peaks corresponding to reduction tend to shift to higher voltage regions while the oxidation peak moves toward lower potentials which reduces the voltage gap from 0.668/0.322 V for pristine samples to 0.504/0.212 V in 3LCVO-ABR”. The above descriptions show two different electrochemical reaction mechanisms, insertion or conversion type? The authors should provide more reasonable explanations and supply more characterizations for verifying the electrochemical reaction mechanism.

Response: We thank Reviewer for this question. We have to apologize for our statement leading to misunderstanding of Reviewer. Herein, the 0.668/0.322 and 0.504/0.212 V are not present for CV peak position but for the different voltage gap between reduction and oxidation peak, as summarized in Table 5. For the further experiment to clarify the electrochemical reaction, we have conducted the *in situ* XRD, as mentioned in response of comment 3, and *ex situ* XPS, as shown in Figure S24. Herein, the *ex situ* XPS is carried out on 3LCVO-ABR electrode at initial state and at stopped potentials of 0.1 and 3.0 V. Accordingly, after discharged to 0.1 V, vs. Li/Li^+ , the V2p XPS consists of three components corresponding to V^{3+} at 215.51 eV, V^{4+} at 516.45 eV, and V^{5+} at 517.42 eV while the V2p XPS at 3.0 V perform a reversible oxidation to V^{4+} and V^{5+} . This

observation is consistent to the conclusion on CV analysis. We have supplied this additional results and discussion as below.

The *ex situ* XPS (Figure S19) was conducted on 3LCVO-ABR electrode at initial state and after being discharged to 0.1 V and charged to 3.0 V, vs. Li/Li⁺. Accordingly, the V2p XPS of 3LCVO-ABR electrode at 0.1 V consists of three components accounted for V³⁺ at 515.51 eV, V⁴⁺ at 516.45 eV, and V⁵⁺ at 517.42 eV. After charged to 3.0 V, vs. Li/Li⁺, the V2p XPS is only composed of V⁴⁺ and V⁵⁺ signals. These results are consistent to the observation in CV.

(Revised manuscript, page 19)

Comment 7: The cycling performance of Li₃VO₄ anode at much higher rate (10C) and longer cycles (>1000) was frequently reported previously. However, the cycling performance in present work is only characterized at a much lower rate of 100mA/g and limited cycles of 200. What the high-rate and long-cycle cycling performance of the studies samples.

Response: Thank Reviewer for suggestion. According to recommendation of Reviewer, we have conducted the experiment on cycling 3LCVO-ABR electrode at higher current density of 1000 (2.5C) and 4000 (10C) mA.g⁻¹. The obtained cycling performance was present in Figure S15. As results, after 1000 cycles, 3LCVO-ABR still offer a high reversible capacity of 477.1 and 337.2 mAh.g⁻¹ with retention of 91.7% and 90.6% at 1000 and 4000 mA.g⁻¹, respectively. The discussion on this additional data was added as below.

Even at a higher current density of 1000 and 4000 mA.g⁻¹ (Figure S13), 3LCVO-ABR still displayed a high reversible capacity of 477.1 and 337.2 mAh.g⁻¹ and capacity retention of 91.7% and 90.6% after 1000 cycles.

(Revised manuscript, page 15)

Comment 8: The unit “mAh/g” has been written as “mA/g” in several places.

Response: We thank Reviewer for this comment. We have carefully checked up and corrected in revised manuscript.

Reviewer 3

Overall comment: In this manuscript, authors developed acid-base reactions to fabricate Li_3VO_4 with controllable morphology and particles size. In addition, a green combination of the acid-base reactions strategy and Ca doping was employed to enhance the electrochemical properties of Li_3VO_4 . This work is significant in green synthesis of Li_3VO_4 anode, and optimization mechanism of Li_3VO_4 . However, the Ca doping Li_3VO_4 in this work is less competitive compared with other Li anode materials. Therefore, I can't recommend this paper to publish in Nature Communications. The following are some comments:

We appreciate Reviewer for spending time to review our manuscript with helpful comment. We have tried to upgrade our work by additional experimental and theoretical calculation results. We hope that our revised manuscript could satisfy Reviewer's requirement.

Comment 1: In recent years, various novel lithium ion battery anode materials have been extensively reported. Many of them have been close to practical applications. According to the overall performance of Ca doping Li_3VO_4 in this work, it's still far away from realizing practical applications.

Response: We thank Reviewer for the comment. We agree with Reviewer that, on the point of view of practical application, our materials' performance is still insufficient. Therefore, to examine the electrochemical properties of our materials at extreme condition, we conducted the experiments on cycling 3LCVO-ABR at higher current density of 1000 and 4000 $\text{mA}\cdot\text{g}^{-1}$ (corresponding to 2.5C and 10C). The obtained results were present in Figure S13. Accordingly, after 1000 cycles, 3LCVO-ABR still delivers a specific capacity of 477.1 and 337.2 $\text{mAh}\cdot\text{g}^{-1}$ with retention of 91.7% and 90.6% at 1000 and 4000 $\text{mA}\cdot\text{g}^{-1}$, respectively. This result indicates the positive effect of Ca-doping and green synthesis strategy. Although the capacity provided by our work can not compete

the state of the art anode reported in recent references, Ca-doped Li_3VO_4 still a good option for anode of lithium ion batteries due to its low cost, and highly stable cycling performance.

Comment 2: The TEM mappings of 3LCVO-ABR should be provided.

Response: We thank Reviewer for suggestion. According to Reviewer's request, we have added the TEM mapping data as shown in Figure S5.

Comment 3: Although the synthesis method is green and facile in this work, it is less competitive compared with other methods such as solvent-free synthesis of silicon-based anode materials.

Response: We thank Reviewer for comment. We agree with Reviewer that the solvent-free synthesis of Si-based materials is a novel strategy for fabricating eco-friendly anodes for lithium ion batteries. However, based on our knowledge, this method is accompanied with high temperature treatment as a magnesiothermal process [Electrochim. Acta, 2020, 352, 136457] which will lead to huge consumption of energy and require high technique to conduct. Therefore, compared to this method, our synthesis conducted at low temperature still possesses its own advantage. Furthermore, as shown in Table S6, the large number of materials which are active to apply as electrodes for lithium ions and sodium ion batteries could demonstrate that our synthesis could be a good candidate for research and practical application.

Comment 4: The Li_3VO_4 with different doping Ca contents possess different morphologies and structures, which may also affect the electrochemical performance.

Response: We thank Reviewer for the recommendation. Accordingly, we have provided the SEM images of 5LCVO-ABR at the same scale compared to 3LCVO-ABR. As shown in Figure R4, there is no significant change in morphology and particle size between 3LCVO-ABR and 5LCVO-ABR. However, the results derived from Rietveld refinement indicate a slight increase of lattice parameters when increasing the doping concentration from 3% to 5%.

Figure R4. SEM images of (a) 3LCVO-ABR; and (b) 5LCVO-ABR.

Comment 5: A specific capacity of 543.1 mA h g⁻¹ at 100 mA g⁻¹ is not superior enough as many works have reported the same performance of Li₃VO₄ anode.

Response: We thank to Reviewer for comment. As mention in previous response, we have conducted a cycling examination of 3LCVO-ABR electrode at higher current density of 1000 and 4000 mA.g⁻¹. After 1000 cycles, the electrode delivers a reversible capacity of 477.1 and 337.2 mAh.g⁻¹ with retention of 91.7% and 90.6% at 1000 and 4000 mA.g⁻¹, respectively. Furthermore, in order to compare to other reported Li₃VO₄-based anodes, we have briefly summarized in Table S13. Accordingly, the electrochemical performance of 3LCVO-ABR in our work are comparable to the other references.

Comment 6: Decreasing the size of Li₃VO₄ particles is a regular method to improving its lithium storage performance, so it's hard to find the highlight of this article.

Response: We thank Reviewer for comment. We agree with Reviewer on the comment that reducing particles size of Li₃VO₄ is a common strategy to improve its electrochemical performance. However, beside our effort in controlling morphology and particles size, in this work, we applied a facile and green method for synthesis a high performance anode for lithium ion batteries. In

addition, for the first time, we proposed a reasonable mechanism of reaction occurred in this synthesis. Besides, the effect of Ca-doping on enhancement in electronic and ionic conductivity was clarified using DFT calculation. Therefore, beside the purpose to fabricate a stable electrode materials which can deliver an acceptable capacity, our work could provide new insights in materials design and synthesis engineering.

1. (a) Shvets, P.; Dikaya, O.; Maksimova, K.; Goikhman, A., A review of Raman spectroscopy of vanadium oxides. *Journal of Raman Spectroscopy* **2019**, *50* (8), 1226-1244; (b) Baddour-Hadjean, R.; Smirnov, M. B.; Kazimirov, V. Y.; Smirnov, K. S.; Pereira-Ramos, J. P., The Raman spectrum of the γ' -V₂O₅ polymorph: a combined experimental and DFT study. *Journal of Raman Spectroscopy* **2015**, *46* (4), 406-412.
2. Qin, R.; Shao, G.; Hou, J.; Zheng, Z.; Zhai, T.; Li, H., One-pot synthesis of Li₃VO₄ @C nanofibers by electrospinning with enhanced electrochemical performance for lithium-ion batteries. *Science Bulletin* **2017**, *62* (15), 1081-1088.
3. Zelang Jian; Mingbo Zheng; Yanliang Liang; Xiaoxue Zhang; Saman Gheyhani; Yucheng Lan; Yi Shib; Yao, Y., Li₃VO₄ anchored graphene nanosheets for long-life and high-rate lithium-ion batteries. *Chem. Commun.* **2015**, *51*, 229.
4. Shi, Y.; Wang, J. Z.; Chou, S. L.; Wexler, D.; Li, H. J.; Ozawa, K.; Liu, H. K.; Wu, Y. P., Hollow structured Li₃VO₄ wrapped with graphene nanosheets in situ prepared by a one-pot template-free method as an anode for lithium-ion batteries. *Nano letters* **2013**, *13* (10), 4715-20.
5. (a) Wang, F.; Liu, Z.; Yuan, X.; Mo, J.; Li, C.; Fu, L.; Zhu, Y.; Wu, X.; Wu, Y., A quasi-solid-state Li-ion capacitor with high energy density based on Li₃VO₄/carbon nanofibers and electrochemically-exfoliated graphene sheets. *Journal of Materials Chemistry A* **2017**, *5* (28), 14922-14929; (b) Shen, L.; Lv, H.; Chen, S.; Kopold, P.; van Aken, P. A.; Wu, X.; Maier, J.; Yu, Y., Peapod-like Li₃VO₄/N-doped carbon nanowires with pseudocapacitive properties as advanced materials for high-energy lithium-ion capacitors. *Advanced Materials* **2017**, *29* (27), 1700142.
6. (a) Yang, S.; Yan, B.; Lu, L.; Zeng, K., Grain boundary effects on Li-ion diffusion in a Li_{1.2}Co_{0.13}Ni_{0.13}Mn_{0.54}O₂ thin film cathode studied by scanning probe microscopy techniques. *RSC Advances* **2016**, *6* (96), 94000-94009; (b) Park, M.; Zhang, X.; Chung, M.; Less, G. B.; Sastry, A. M., A review of conduction phenomena in Li-ion batteries. *Journal of Power Sources* **2010**, *195* (24), 7904-7929.
7. Lee, E.-J.; Chen, Z.; Noh, H.-J.; Nam, S. C.; Kang, S.; Kim, D. H.; Amine, K.; Sun, Y.-K., Development of microstrain in aged lithium transition metal oxides. *Nano letters* **2014**, *14* (8), 4873-4880.

Reviewer #1 (Remarks to the Author):

I have carefully evaluated all response from the authors, they have provided the satisfactory revision. The paper can be accepted in current form.

Reviewer #3 (Remarks to the Author):

After the revision, the authors have revised the manuscript according to the comments earnestly, and I think that this manuscript can be received by Nature Communications. I believe this article titled "Sub-micro droplet reactors: Green synthesis of Li_3VO_4 anode materials for lithium ion batteries" will be of high interest to the broad readers in the material and energy fields.

Reviewer 1

Overall comment: I have carefully evaluated all response from the authors, they have provided the satisfactory revision. The paper can be accepted in current form.

We appreciate the Reviewer's valuable evaluation and acceptance for our publication on *Nature Communications*.

Reviewer 3

Overall comment: After the revision, the authors have revised the manuscript according to the comments earnestly, and I think that this manuscript can be received by Nature Communications. I believe this article titled "Sub-micro droplet reactors: Green synthesis of Li_3VO_4 anode materials for lithium ion batteries" will be of high interest to the broad readers in the material and energy fields.

We appreciate Reviewer for spending time to review our manuscript and recommendation for our publication on *Nature Communications*.